# SPIKFORMER: WHEN SPIKING NEURAL NETWORK MEETS TRANSFORMER

[1,2]**Zhaokun Zhou**   [1,2]**Yuesheng Zhu**[*]   [4]**Chao He**   [2]**Yaowei Wang**   [3]**Shuicheng Yan**
[1,2]**Yonghong Tian**   [1,2]**Li Yuan**[*]
[1]Peking University   [2]Peng Cheng Laboratory   [3]Sea AI Lab
[4] Shenzhen EEGSmart Technology Co., Ltd.
{yuanli-ece}@pku.edu.cn

## ABSTRACT

We consider two biologically plausible structures, the Spiking Neural Network (SNN) and the self-attention mechanism. The former offers an energy-efficient and event-driven paradigm for deep learning, while the latter has the ability to capture feature dependencies, enabling Transformer to achieve good performance. It is intuitively promising to explore the marriage between them. In this paper, we consider leveraging both self-attention capability and biological properties of SNNs, and propose a novel Spiking Self Attention (SSA) as well as a powerful framework, named Spiking Transformer (Spikformer). The SSA mechanism in Spikformer models the sparse visual feature by using spike-form Query, Key, and Value without softmax. Since its computation is sparse and avoids multiplication, SSA is efficient and has low computational energy consumption. It is shown that Spikformer with SSA can outperform the state-of-the-art SNNs-like frameworks in image classification on both neuromorphic and static datasets. Spikformer (66.3M parameters) with comparable size to SEW-ResNet-152 (60.2M, 69.26%) can achieve 74.81% top1 accuracy on ImageNet using 4 time steps, which is the state-of-the-art in directly trained SNNs models. Codes is avaiable at Spikformer.

## 1 INTRODUCTION

As the third generation of neural network (Maass, 1997), the Spiking Neural Network (SNN) is very promising for its low power consumption, event-driven characteristic, and biological plausibility (Roy et al., 2019). With the development of artificial neural networks (ANNs), SNNs are able to lift performance by borrowing advanced architectures from ANNs, such as ResNet-like SNNs (Hu et al., 2021a; Fang et al., 2021a; Zheng et al., 2021; Hu et al., 2021b), Spiking Recurrent Neural Networks (Lotfi Rezaabad & Vishwanath, 2020) and Spiking Graph Neural Networks (Zhu et al., 2022). Transformer, originally designed for natural language processing (Vaswani et al., 2017), has flourished for various tasks in computer vision, including image classification (Dosovitskiy et al., 2020; Yuan et al., 2021a), object detection (Carion et al., 2020; Zhu et al., 2020; Liu et al., 2021), semantic segmentation (Wang et al., 2021; Yuan et al., 2021b) and low-level image processing (Chen et al., 2021). Self-attention, the key part of Transformer, selectively focuses on information of interest, and is also an important feature of the human biological system (Whittington et al., 2022; Caucheteux & King, 2022). Intuitively, it is intriguing to explore applying self-attention in SNNs for more advanced deep learning, considering the biological properties of the two mechanisms.

It is however non-trivial to port the self-attention mechanism into SNNs. In vanilla self-attention (VSA) (Vaswani et al., 2017), there are three components: Query, Key, and Value. As shown in Figure 1(a), standard inference of VSA is firstly obtaining a matrix by computing the dot product of float-point-form Query and Key; then softmax, which contains exponential calculations and division operations, is adopted to normalize the matrix to give the attention map which will be used to weigh the Value. The above steps in VSA do not conform to the calculation characteristics of SNNs, i.e., avoiding multiplication. Moreover, the heavy computational overhead of VSA almost prohibits

---

[*]Indicates the corresponding author.

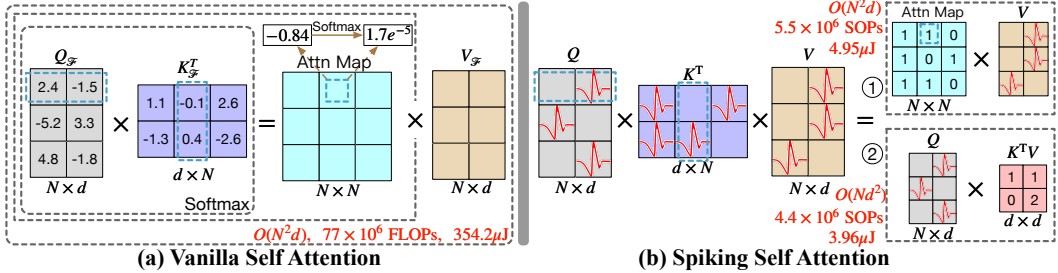

Figure 1: Illustration of vanilla self-attention (VSA) and our Spiking Self Attention (SSA). A red spike indicates a value of 1 at that location. The blue dashed boxes provide examples of matrix dot product operation. For convenience, we choose one of the heads of SSA, where $N$ is the number of input patches and $d$ is the feature dimension of one head. FLOPs is the floating point operations and SOPs is the theoretical synaptic operations. The theoretical energy consumption to perform one calculation between Query, Key and Value in one time step is obtained from 8-encoder-blocks 512-embedding-dimension Spikformer on ImageNet test set according to (Kundu et al., 2021b; Hu et al., 2021a). More details about the calculation of theoretical SOP and energy consumption are included in appendix. C.2. (a) In VSA, $Q_\mathcal{F}, K_\mathcal{F}, V_\mathcal{F}$ are float-point forms. After the dot-product of $Q_\mathcal{F}$ and $K_\mathcal{F}$, the softmax function regularizes negative values in the attention map to positive values. (b) In SSA, all value in attention map is non-negative and the computation is sparse using spike-form $Q, K, V$ ($5.5 \times 10^6$ VS. $77 \times 10^6$ in VSA). Therefore, the computation in SSA consumes less energy compared with VSA ($354.2\mu$J). In addition, the SSA is decomposable (the calculation order of $Q, K$ and $V$ is changeable).

applying it directly to SNNs. Therefore, in order to develop Transformer on SNNs, we need to design a new effective and computation-efficient self-attention variant that can avoid multiplications.

We thus present Spiking Self Attention (SSA), as illustrated in Figure 1(b). SSA introduces self-attention mechanism to SNNs for the first time, which models the interdependence using spike sequences. In SSA, the Query, Key, and Value are in spike form which only contains of 0 and 1. The obstacles to the application of self-attention in SNNs are mainly caused by softmax. 1) As shown in Figure 1, the attention map calculated from spike-form Query and Key has natural non-negativeness, which ignores irrelevant features. Thus, we do not need the softmax to keep the attention matrix non-negative, which is its most important role in VSA (Qin et al., 2022). 2) The input and the Value of the SSA are in the form of spikes, which only consist of 0 and 1 and contain less fine-grained feature compared to the float-point input and Value of the VSA in ANNs. So the float-point Query and Key and softmax function are redundant for modeling such spike sequences. Tab. 1 illustrates that our SSA is competitive with VSA in the effect of processing spike sequences. Based on the above insights, we discard softmax normalization for the attention map in SSA. Some previous Transformer variants also discard softmax or replace it with a linear function. For example, in Performer (Choromanski et al., 2020), positive random feature is adopted to approximate softmax; CosFormer (Qin et al., 2022) replaces softmax with ReLU and cosine function.

With such designs of SSA, the calculation of spike-form Query, Key, and Value avoids multiplications and can be done by logical AND operation and addition. Also, its computation is very efficient. Due to sparse spike-form Query, Key and Value (shown in appendix D.1) and simple computation, the number of operations in SSA is small, which makes the energy consumption of SSA very low. Moreover, our SSA is decomposable after deprecation of softmax, which further reduces its computational complexity when the sequence length is greater than the feature dimension of one head, as depicted in Figure 1(b) ① ②.

Based on the proposed SSA, which well suits the calculation characteristics of SNNs, we develop the Spiking Transformer (Spikformer). An overview of Spikformer is shown in Figure 2. It boosts the performance trained on both static datasets and neuromorphic datasets. To the best of our knowledge, it is the first time to explore the self-attention mechanism and directly-trained Transformer in the SNNs. To sum up, there are three-fold contributions of our work:

- We design a novel spike-form self-attention named Spiking Self Attention (SSA) for the properties of SNNs. Using sparse spike-form Query, Key, and Value without softmax, the calculation of SSA avoids multiplications and is efficient.
- We develop the Spiking Transformer (Spikformer) based on the proposed SSA. To the best of our knowledge, this is the first time to implement self-attention and Transformer in SNNs.

- Extensive experiments show that the proposed architecture outperforms the state-of-the-art SNNs on both static and neuromorphic datasets. It is worth noting that we achieved more than 74% accuracy on ImageNet with 4 time steps using directly-trained SNN model for the first time.

## 2 RELATED WORK

**Vision Transformers.** For the image classification task, a standard vision transformer (ViT) includes a patch splitting module, the transformer encoder layer(s), and linear classification head. The Transformer encoder layer consists of a self-attention layer and a multi perception layer block. Self-attention is the core component making ViT successful. By weighting the image-patches feature value through the dot-product of query and key and softmax function, self-attention can capture the global dependence and interest representation (Katharopoulos et al., 2020; Qin et al., 2022). Some works have been carried out to improve the structures of ViTs. Using convolution layers for patch splitting has been proven to be able to accelerate convergence and alleviate the data-hungry problem of ViT (Xiao et al., 2021b; Hassani et al., 2021). There are some methods aiming to reduce the computational complexity of self-attention or improve its ability of modeling visual dependencies (Song, 2021; Yang et al., 2021; Rao et al., 2021; Choromanski et al., 2020). This paper focuses on exploring the effectiveness of self-attention in SNNs and developing a powerful spiking transformer model for image classification.

**Spiking Neural Networks.** Unlike traditional deep learning models that convey information using continuous decimal values, SNNs use discrete spike sequences to calculate and transmit information. Spiking neurons receive continuous values and convert them into spike sequences, including the Leaky Integrate-and-Fire (LIF) neuron (Wu et al., 2018), PLIF (Fang et al., 2021b), etc. There are two ways to get deep SNN models: ANN-to-SNN conversion and direct training. In ANN-to-SNN conversion (Cao et al., 2015; Hunsberger & Eliasmith, 2015; Rueckauer et al., 2017; Bu et al., 2021; Meng et al., 2022; Wang et al., 2022), the high-performance pre-trained ANN is converted to SNN by replacing the ReLU activation layers with spiking neurons. The converted SNN requires large time steps to accurately approximate ReLU activation, which causes large latency (Han et al., 2020). In the area of direct training, SNNs are unfolded over the simulation time steps and trained in a way of backpropagation through time (Lee et al., 2016; Shrestha & Orchard, 2018). Because the event-triggered mechanism in spiking neurons is non-differentiable, the surrogate gradient is used for backpropagation (Lee et al., 2020; Neftci et al., 2019)Xiao et al. (2021a) adopts implicit differentiation on the equilibrium state to train SNN. Various models from ANNs have been ported to SNNs. However, the study of self-attention on SNN is currently blank. Yao et al. (2021) proposed temporal attention to reduce the redundant time step. Zhang et al. (2022a;b) both use ANN-Transformer to process spike data, although they have 'Spiking Transformer' in the title. Mueller et al. (2021) provides a ANN-SNN conversion Transformer, but remains vanilla self-attention which does not conform the characteristic of SNN. In this paper, we will explore the feasibility of implementing self-attention and Transformer in SNNs.

As the fundamental unit of SNNs, the spike neuron receives the resultant current and accumulates membrane potential which is used to compare with the threshold to determine whether to generate the spike. We uniformly use LIF spike neurons in our work. The dynamic model of LIF is described as:

$$H[t] = V[t-1] + \frac{1}{\tau}\left(X[t] - (V[t-1] - V_{reset})\right), \tag{1}$$

$$S[t] = \Theta(H[t] - V_{th}), \tag{2}$$

$$V[t] = H[t]\left(1 - S[t]\right) + V_{reset}S[t], \tag{3}$$

where $\tau$ is the membrane time constant, and $X[t]$ is the input current at time step $t$. When the membrane potential $H[t]$ exceeds the firing threshold $V_{th}$, the spike neuron will trigger a spike $S[t]$. $\Theta(v)$ is the Heaviside step function which equals 1 for $v \geq 0$ and 0 otherwise. $V[t]$ represents the membrane potential after the trigger event which equals $H[t]$ if no spike is generated, and otherwise equals to the reset potential $V_{reset}$.

## 3 METHOD

We propose Spiking Transformer (Spikformer), which incorporates the self-attention mechanism and Transformer into the spiking neural networks (SNNs) for enhanced learning capability. Now we explain the overview and components of Spikformer one by one.

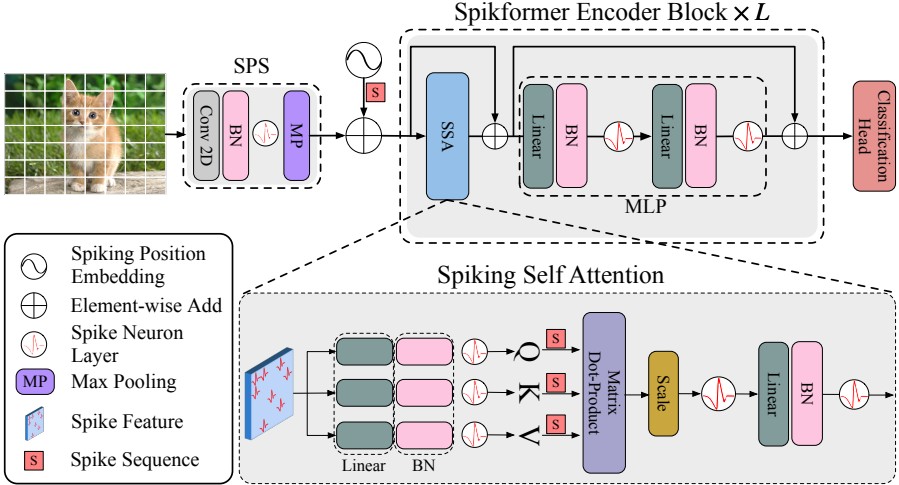

Figure 2: The overview of Spiking Transformer (Spikformer), which consists of a spiking patch splitting module (SPS), a Spikformer encoder and a Linear classification head. We empirically find that the layer normalization (LN) does not apply to SNNs, so we use batch normalization (BN) instead.

## 3.1 OVERALL ARCHITECTURE

An overview of Spikformer is depicted in Figure 2. Given a 2D image sequence $I \in \mathbb{R}^{T \times C \times H \times W}$[1], the Spiking Patch Splitting (SPS) module linearly projects it to a $D$ dimensional spike-form feature vector and splits it into a sequence of $N$ flattened spike-form patches $x$. Float-point-form position embedding cannot be used in SNNs. We employ a conditional position embedding generator (Chu et al., 2021) to generate spike-form relative position embedding (RPE) and add the RPE to patches sequence $x$ to get $X_0$. The conditional position embedding generator contains a 2D convolution layer (Conv2d) with kernel size 3, batch normalization (BN), and spike neuron layer ($\mathcal{SN}$). Then we pass the $X_0$ to the $L$-block Spikformer encoder. Similar to the standard ViT encoder block, a Spikformer encoder block consists of a Spiking Self Attention (SSA) and an MLP block. Residual connections are applied in both the SSA and MLP block. As the main component in Spikformer encoder block, SSA offers an efficient method to model the local-global information of images using spike-form Query ($Q$), Key ($K$), and Value ($V$) without softmax, which will be analyzed in detail in Sec. 3.3. A global average-pooling (GAP) is utilized on the processed feature from Spikformer encoder and outputs the $D$-dimension feature which will be sent to the fully-connected-layer classification head (CH) to output the prediction $Y$. Spikformer can be written as follows:

$$x = \text{SPS}(I), \qquad\qquad I \in \mathbb{R}^{T \times C \times H \times W}, x \in \mathbb{R}^{T \times N \times D}, \qquad (4)$$

$$\text{RPE} = \mathcal{SN}(\text{BN}((\text{Conv2d}(x)))), \qquad \text{RPE} \in \mathbb{R}^{T \times N \times D} \qquad (5)$$

$$X_0 = x + \text{RPE}, \qquad\qquad X_0 \in \mathbb{R}^{T \times N \times D} \qquad (6)$$

$$X'_l = \text{SSA}(X_{l-1}) + X_{l-1}, \qquad X'_l \in \mathbb{R}^{T \times N \times D}, l = 1...L \qquad (7)$$

$$X_l = \text{MLP}(X'_l) + X'_l, \qquad\qquad X_l \in \mathbb{R}^{T \times N \times D}, l = 1...L \qquad (8)$$

$$Y = \text{CH}(\text{GAP}(X_L)) \qquad\qquad (9)$$

## 3.2 SPIKING PATCH SPLITTING

As shown in Figure 2, the Spiking Patch Splitting (SPS) module aims to linearly project an image to a $D$ dimensional spike-form feature and split the feature into patches with a fixed size. SPS can contain multiple blocks. Similar to the convolutional stem in Vision Transformer (Xiao et al., 2021b; Hassani et al., 2021), we apply a convolution layer in each SPS block to introduce inductive bias into

---

[1]In the neuromorphic dataset the data shape is $I \in \mathbb{R}^{T \times C \times H \times W}$, where $T$, $C$, $H$, and $W$ denote time step, channel, height and width, respectively. A 2D image $I_s \in \mathbb{R}^{C \times H \times W}$ in static datasets need to be repeated $T$ times to form a sequence of images.

Spikformer. Specifically, given an image sequence $I \in \mathbb{R}^{T \times C \times H \times W}$:

$$x = \mathcal{MP}\left(\mathcal{SN}(\mathrm{BN}((\mathrm{Conv2d}(I))))\right) \tag{10}$$

where the Conv2d and $\mathcal{MP}$ represent the 2D convolution layer (stride-1, $3 \times 3$ kernel size) and max-pooling, respectively. The number of SPS blocks can be more than 1. When using multiple SPS blocks, the number of output channels in these convolution layers is gradually increased and finally matches the embedding dimension of patches. For example, given an output embedding dimension $D$ and a four-block SPS module, the number of output channels in four convolution layers is $D/8, D/4, D/2, D$. While the 2D-max-pooling layer is applied to down-sample the feature size after SPS block with a fixed size. After the processing of SPS, $I$ is split into an image patches sequence $x \in \mathbb{R}^{T \times N \times D}$.

### 3.3 Spiking Self Attention Mechanism

Spikformer encoder is the main component of the whole architecture, which contains the Spiking Self Attention (SSA) mechanism and MLP block. In this section we focus on SSA, starting with a review of vanilla self-attention (VSA). Given an input feature sequence $X \in \mathbb{R}^{T \times N \times D}$, the VSA in ViT has three float-point key components, namely query ($Q_{\mathcal{F}}$), key ($K_{\mathcal{F}}$), and value ($V_{\mathcal{F}}$) which are calculated by learnable linear matrices $W_Q, W_K, W_V \in \mathbb{R}^{D \times D}$ and $X$:

$$Q_{\mathcal{F}} = XW_Q, \ K_{\mathcal{F}} = XW_K, \ V_{\mathcal{F}} = XW_V \tag{11}$$

where $\mathcal{F}$ denotes the float-point form. The output of vanilla self-attention can be computed as:

$$\mathrm{VSA}(Q_{\mathcal{F}}, K_{\mathcal{F}}, V_{\mathcal{F}}) = \mathrm{Softmax}\left(\frac{Q_{\mathcal{F}} K_{\mathcal{F}}^{\mathrm{T}}}{\sqrt{d}}\right) V_{\mathcal{F}} \tag{12}$$

where $d = D/H$ is the feature dimension of one head and $H$ is the head number. Converting the float-point-form Value ($V_{\mathcal{F}}$) into spike form ($V$) can realize the direct application of VSA in SNNs, which can be expressed as:

$$\mathrm{VSA}(Q_{\mathcal{F}}, K_{\mathcal{F}}, V) = \mathrm{Softmax}\left(\frac{Q_{\mathcal{F}} K_{\mathcal{F}}^{\mathrm{T}}}{\sqrt{d}}\right) V \tag{13}$$

However, the calculation of VSA is not applicable in SNNs for two reasons. 1) The float-point matrix multiplication of $Q_{\mathcal{F}}, K_{\mathcal{F}}$ and softmax function which contains exponent calculation and division operation, do not comply with the calculation rules of SNNs. 2) The quadratic space and time complexity of the sequence length of VSA do not meet the efficient computational requirements of SNNs.

We propose Spiking Self-Attention (SSA), which is more suitable for SNNs than the VSA, as shown in Figure 1(b) and the bottom of Figure 2. The query ($Q$), key ($K$), and Value ($V$) are computed through learnable matrices firstly. Then they become spiking sequences via different spike neuron layers:

$$Q = \mathcal{SN}_Q(\mathrm{BN}(XW_Q)), K = \mathcal{SN}_K(\mathrm{BN}(XW_K)), V = \mathcal{SN}_V(\mathrm{BN}(XW_V)) \tag{14}$$

where $Q, K, V \in \mathbb{R}^{T \times N \times D}$. We believe that the calculation process of the attention matrix should use pure spike-form Query and Key(only containing 0 and 1). Inspired by vanilla self-attention (Vaswani et al., 2017), we add a scaling factor $s$ to control the large value of the matrix multiplication result. $s$ does not affect the property of SSA. As shown in Figure 2, the spike-friendly SSA is defined as:

$$\mathrm{SSA}'(Q, K, V) = \mathcal{SN}\left(Q \ K^{\mathrm{T}} \ V * s\right) \tag{15}$$

$$\mathrm{SSA}(Q, K, V) = \mathcal{SN}(\mathrm{BN}(\mathrm{Linear}(\mathrm{SSA}'(Q, K, V)))). \tag{16}$$

The single-head SSA introduced here can easily be extended to the multi-head SSA, which is detailed in the appendix A. SSA is independently conducted on each time step and seeing more details in appendix B. As shown in Eq. (15), SSA cancels the use of softmax to normalize the attention matrix in Eq. (12) and directly multiplies $Q, K$ and $V$. An intuitive calculation example is shown in Figure 1(b). The softmax is unnecessary in our SSA, and it even hinders the implementation of self-attention to SNNs. Formally, based on Eq. (14), the spike sequences $Q$ and $K$ output by the spiking neuron

layer $\mathcal{SN}_Q$ and $\mathcal{SN}_k$ respectively, are naturally non-negative (0 or 1), resulting in a non-negative attention map. SSA only aggregates these relevant features and ignores the irrelevant information. Hence it does not need the softmax to ensure the non-negativeness of the attention map. Moreover, compared to the float-point-form $X_{\mathcal{F}}$ and $V_{\mathcal{F}}$ in ANNs, the input $X$ and the Value $V$ of self-attention in SNNs are in spike form, containing limited information. The vanilla self-attention (VSA) with float-point-form $Q_{\mathcal{F}}, K_{\mathcal{F}}$ and softmax is redundant for modeling the spike-form $X, V$, which cannot get more information from $X, V$ than SSA. That is, SSA is more suitable for SNNs than the VSA.

We conduct experiments to validate the above insights by comparing the proposed SSA with four different calculation methods of the attention map, as shown in Tab. 1. $A_I$ denotes multiplying the float-points $Q$ and $K$ directly to get the attention map, which preserves both positive and negative correlation. $A_{\text{ReLU}}$ uses the multiplication between $\text{ReLU}(Q)$ and $\text{ReLU}(K)$ to obtain the attention map. $A_{\text{ReLU}}$ retains the positive values of $Q, K$ and sets the negative values to 0,

Table 1: Analysis of the SSA's rationality. We replace SSA with other attention variants and keep the remaining network structure in Spikformer unchanged. We show the accuracy (Acc) on CIFAR10-DVS (Li et al., 2017), CIFAR10/100 (Krizhevsky, 2009). OPs (M) is the number of operations (For $A_I, A_{\text{LeakyReLU}}, A_{\text{ReLU}}$ and $A_{\text{softmax}}$, OPs is FLOPs, and SOPs is ignored; For $A_{\text{SSA}}$, it is SOPs.) and P ($\mu$J) is the theoretical energy consumption to perform one calculation among $Q, K, V$.

| | CIFAR10-DVS | CIFAR10 | CIFAR100 |
|---|---|---|---|
| | Acc/OPs (M)/P ($\mu$J) | | |
| $A_I$ | 79.40/16.8/77 | 93.96/6.3/29 | 76.94/6.3/29 |
| $A_{\text{LeakyReLU}}$ | 79.80/16.8/77 | 93.85/6.3/29 | 76.73/6.3/29 |
| $A_{\text{ReLU}}$ | 79.40/16.8/77 | 94.34/6.3/29 | 77.00/6.3/29 |
| $A_{\text{softmax}}$ | 80.00/19.1/88 | 94.97/6.6/30 | **77.92**/6.6/30 |
| $A_{\text{SSA}}$ | **80.90/0.66/0.594** | **95.19/1.1/0.990** | 77.86/**1.3/1.170** |

while $A_{\text{LeakyReLU}}$ still retains the negative points. $A_{\text{softmax}}$ means the attention map is generated following VSA. The above four methods use the same Spikformer framework and weight the spike-form $V$. From Tab. 1, the superior performance of our $A_{\text{SSA}}$ over $A_I$ and $A_{\text{LeakyReLU}}$ proves the superiority of $\mathcal{SN}$. The reason why $A_{\text{SSA}}$ is better than $A_{\text{ReLU}}$ may be that $A_{\text{SSA}}$ has better non-linearity in self-attention. By comparing with $A_{\text{softmax}}$, $A_{\text{SSA}}$ is competitive, which even surpasses $A_{\text{softmax}}$ on CIFAR10DVS and CIFAR10. This can be attributed to SSA being more suitable for spike sequences ($X$ and $V$) with limited information than VSA. Furthermore, the number of operations and theoretical energy consumption required by the $A_{\text{SSA}}$ to complete the calculation of $Q, K, V$ is much lower than that of the other methods.

SSA is specially designed for modeling spike sequences. The $Q, K$, and $V$ are all in spike form, which degrades the matrix dot-product calculation to logical AND operation and summation operation. We take a row of Query $q$ and a column of Key $k$ as a calculation example: $\sum_{i=1}^{d} q_i k_i = \sum_{q_i=1} k_i$. Also, as shown in Tab. 1, SSA has a low computation burden and energy consumption due to sparse spike-form $Q, K$ and $V$ (Figure. 4) and simplified calculation. In addition, the order of calculation between $Q, K$ and $V$ is changeable: $QK^{\text{T}}$ first and then $V$, or $K^{\text{T}}V$ first and then $Q$. When the sequence length $N$ is bigger than one head dimension $d$, the second calculation order above will incur less computation complexity ($O(Nd^2)$) than the first one ($O(N^2d)$). SSA maintains the biological plausibility and computationally efficient properties throughout the whole calculation process.

## 4 EXPERIMENTS

We conduct experiments on both static datasets CIFAR, ImageNet (Deng et al., 2009), and neuro-morphic datasets CIFAR10-DVS, DVS128 Gesture (Amir et al., 2017) to evaluate the performance of Spikformer. The models for conducting experiments are implemented based on Pytorch (Paszke et al., 2019), SpikingJelly [2] and Pytorch image models library (Timm) [3]. We train the Spikformer from scratch and compare it with current SNNs models in Sec. 4.1 and 4.2. We conduct ablation studies to show the effects of the SSA module and Spikformer in Sec. 4.3.

### 4.1 STATIC DATASETS CLASSIFICATION

**ImageNet** contains around $1.3$ million $1,000$-class images for training and $50,000$ images for validation. The input size of our model on ImageNet is set to the default $224 \times 224$. The optimizer

---
[2] https://github.com/fangwei123456/spikingjelly
[3] https://github.com/rwightman/pytorch-image-models

Table 2: Evaluation on ImageNet. Param refers to the number of parameters. Power is the average theoretical energy consumption when predicting an image from ImageNet test set, whose calculation detail is shown in Eq. 22. Spikformer-$L$-$D$ represents a Spikformer model with $L$ Spikformer encoder blocks and $D$ feature embedding dimensions. The train loss, test loss and test accuracy curves are shown in appendix D.2. OPs refers to SOPs in SNN and FLOPs in ANN-ViT.

| Methods | Architecture | Param (M) | OPs (G) | Power (mJ) | Time Step | Acc |
|---|---|---|---|---|---|---|
| Hybrid training(Rathi et al., 2020) | ResNet-34 | 21.79 | - | - | 250 | 61.48 |
| TET(Deng et al., 2021) | Spiking-ResNet-34 | 21.79 | - | - | 6 | 64.79 |
| | SEW-ResNet-34 | 21.79 | - | - | 4 | 68.00 |
| Spiking ResNet(Hu et al., 2021a) | ResNet-34 | 21.79 | 65.28 | 59.295 | 350 | 71.61 |
| | ResNet-50 | 25.56 | 78.29 | 70.934 | 350 | 72.75 |
| STBP-tdBN(Zheng et al., 2021) | Spiking-ResNet-34 | 21.79 | 6.50 | 6.393 | 6 | 63.72 |
| | SEW-ResNet-34 | 21.79 | 3.88 | 4.035 | 4 | 67.04 |
| SEW ResNet(Fang et al., 2021a) | SEW-ResNet-50 | 25.56 | 4.83 | 4.890 | 4 | 67.78 |
| | SEW-ResNet-101 | 44.55 | 9.30 | 8.913 | 4 | 68.76 |
| | SEW-ResNet-152 | 60.19 | 13.72 | 12.891 | 4 | 69.26 |
| Transformer | Transformer-8-512 | 29.68 | 8.33 | 38.340 | 1 | **80.80** |
| **Spikformer** | Spikformer-8-384 | 16.81 | 6.82 | 7.734 | 4 | 70.24 |
| | Spikformer-6-512 | 23.37 | 8.69 | 9.417 | 4 | 72.46 |
| | Spikformer-8-512 | 29.68 | 11.09 | 11.577 | 4 | **73.38** |
| | Spikformer-10-512 | 36.01 | 13.67 | 13.899 | 4 | **73.68** |
| | Spikformer-8-768 | 66.34 | 22.09 | 21.477 | 4 | **74.81** |

is AdamW and the batch size is set to 128 or 256 during 310 training epochs with a cosine-decay learning rate whose initial value is 0.0005. The scaling factor is 0.125 when training on ImageNet and CIFAR. A four-block SPS splits the image into 196 $16 \times 16$ patches. Following (Yuan et al., 2021a), standard data augmentation methods, such as random augmentation, mixup, and cutmix, are also used in training. We try a variety of models with different embedding dimensions and numbers of transformer blocks for ImageNet, which has been shown in Tab. 2. We also give a comparison of synaptic operations (SOPs) (Merolla et al., 2014) and theoretical energy consumption.

From the results, it can be seen that our Spikformer achieves a significant accuracy boost on the ImageNet compared with the current best SNNs models. In particular, our comparison first starts from our smallest model with other models. The Spikformer-8-384 with 16.81M parameters has 70.24% top-1 accuracy when trained from scratch on ImageNet, which outperforms the best the current best direct-train model SEW-ResNet-152: 69.26% with 60.19M. In addition, the SOPs and the theoretical energy consumption of Spikformer-8-384 (6.82G, 7.734mJ) are lower compared with the SEW-ResNet-152 (13.72G, 12.891mJ). The 29.68M model Spikformer-8-512 has already achieved state-of-the-art performance with 73.38%, which is even higher than the converted model (Hu et al., 2021a) (72.75%) using 350 time steps. As the number of Spikformer blocks increases, the classification accuracy of our model on ImageNet is also getting higher. The Spikformer-10-512 obtains 73.68% with 42.35M. The same happens when gradually increasing the embedding dimension, where Spikformer-8-768 further improves the performance to 74.81% and significantly outperforms the SEW-ResNet-152 model by 5.55%. ANN-ViT-8-512 is 7.42% higher than Spikformer-8-512, but the theoretical energy consumption is $3.31\times$ of Spikformer-8-512. In Figure 3, we show the attention map examples of the last encoder block in Spikformer-8-512 at the fourth time step. SSA can capture image regions associated with classification semantics and set irrelevant regions to 0 (black region), and is shown to be effective, event-driven, and energy-efficient.

Input Image     Attention Map

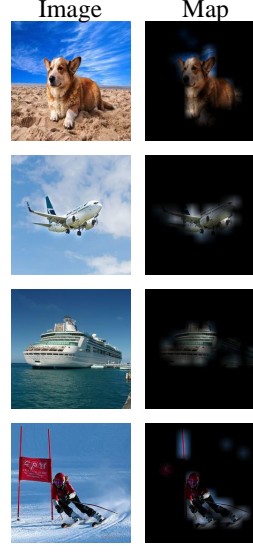

Figure 3: Attention map examples of SSA. The black region is 0.

**CIFAR** provides $50,000$ train and $10,000$ test images with $32 \times 32$ resolution. The batch size is set to 128. A four-block SPS (the first two blocks do not contain the max-pooling layer) splits the image into 64 $4 \times 4$ patches. Tab. 3 shows the accuracy of Spikformer compared with other models on CIFAR. As shown in Tab. 3, Spikformer-4-384 achieves 95.19% accuracy on CIFAR10, which is better than the TET (94.44%) and ResNet-19

Table 3: Performance comparison of our method with existing methods on CIFAR10/100. Our method improves network performance across all tasks. * denotes self-implementation results by Deng et al. (2021). Note that Hybrid training (Rathi et al., 2020) adopts ResNet-20 for CIFAR10 and VGG-11 for CIFAR100.

| Methods | Architecture | Param (M) | Time Step | CIFAR10 Acc | CIFAR100 Acc |
|---|---|---|---|---|---|
| Hybrid training(Rathi et al., 2020) | VGG-11 | 9.27 | 125 | 92.22 | 67.87 |
| Diet-SNN(Rathi & Roy, 2020) | ResNet-20 | 0.27 | 10/5 | 92.54 | 64.07 |
| STBP(Wu et al., 2018) | CIFARNet | 17.54 | 12 | 89.83 | - |
| STBP NeuNorm(Wu et al., 2019) | CIFARNet | 17.54 | 12 | 90.53 | - |
| TSSL-BP(Zhang & Li, 2020) | CIFARNet | 17.54 | 5 | 91.41 | - |
| STBP-tdBN(Zheng et al., 2021) | ResNet-19 | 12.63 | 4 | 92.92 | 70.86 |
| TET(Deng et al., 2021) | ResNet-19 | 12.63 | 4 | **94.44** | **74.47** |
| **ANN** | ResNet-19* | 12.63 | 1 | **94.97** | **75.35** |
| | Transformer-4-384 | 9.32 | 1 | **96.73** | **81.02** |
| **Spikformer** | Spikformer-4-256 | 4.15 | 4 | 93.94 | 75.96 |
| | Spikformer-2-384 | 5.76 | 4 | **94.80** | 76.95 |
| | Spikformer-4-384 | 9.32 | 4 | **95.19** | 77.86 |
| | Spikformer-4-384 400E | 9.32 | 4 | **95.51** | 78.21 |

ANN ($94.97\%$). The performance is improved as the dimensions or blocks increase. Specifically, Spikformer-4-384 improves by $1.25\%$ compared to Spikformer-4-256 and improves by $0.39\%$ compared to Spikformer-2-384. We also find that extending the number of training epochs to 400 can improve the performance (Spikformer-4-384 400E achieves $0.32\%$ and $0.35\%$ advance compared to Spikformer-4-384 on CIFAR10 and CIFAR100). The improvement of the proposed Spikformer on complex datasets such as CIFAR100 is even higher. Spikformer-4-384 ($77.86\%, 9.32M$) obtains a significant improvement of $2.51\%$ compared with ResNet-19 ANN ($75.35\%, 12.63M$) model. The ANN-Transformer model is $1.54\%$ and $3.16\%$ higher than Spikformer-4-384, respectively. As shown in appendix D.5, transfer learning can achieve higher performance on CIFAR based on pre-trained Spikformer, which demonstrates high transfer ability.

## 4.2 NEUROMORPHIC DATASETS CLASSIFICATION

DVS128 Gesture is a gesture recognition dataset that contains 11 hand gesture categories from 29 individuals under 3 illumination conditions. CIFAR10-DVS is also a neuromorphic dataset converted from the static image dataset by shifting image samples to be captured by the DVS camera, which provides $9,000$ training samples and $1,000$ test samples.

For the above two datasets of image size $128 \times 128$, we adopt a four-block SPS. The patch embedding dimension is 256 and the patch size is $16 \times 16$. We use a shallow Spikformer with 2 transformer encoder blocks. The SSA contains 8 and 16 heads for DVS128 Gesture and CIFAR10-DVS, respectively. The time-step of the spiking neuron is 10 or 16. The training epoch is 200 for DVS128 Gesture and 106 for CIFAR10-DVS. The optimizer is AdamW and the batch size is set to 16. The learning rate is initialized to $0.1$ and reduced with cosine decay. We apply data augmentation on CIFAR10-DVS according to (Li et al., 2022). We use a learnable parameter as the scaling factor to control the $QK^\mathrm{T}V$ result.

The classification performance of Spikformer as well as the compared state-of-the-art models on neuromorphic datasets is shown in Tab. 4. It can be seen that our model achieves good performance on both datasets by using a 2.59M model. On DVS128 Gesture, we obtain an accuracy of $98.2\%$ with 16-time steps, which is higher than SEW-ResNet ($97.9\%$). Our result is also competitive compared with TA-SNN ($98.6\%$, 60 time steps) (Yao et al., 2021) which uses floating-point spikes in the forward propagation. On CIFAR10-DVS, we achieve a $1.6\%$ and $3.6\%$ better accuracy than the SOTA methods DSR ($77.3\%$) with binary spikes using 10 steps and 16 steps respectively. TET is not an architecture-based but a loss-based method which achieves $83.2\%$ using long epochs (300) and 9.27M VGGSNN, so we do not compare with it in the table.

## 4.3 ABLATION STUDY

**Time step** The accuracy regarding different simulation time steps of the spike neuron is shown in Tab. 5. When the time step is 1, our method is $1.87\%$ lower than the network with $T = 4$ on CIFAR10.

Table 4: Performance comparison to the state-of-the-art (SOTA) methods on two neuromorphic datasets. Bold font means the best; * denotes with Data Augmentation.

| Method | Spikes | CIFAR10-DVS | | DVS128 | |
|---|---|---|---|---|---|
| | | $T$ Step | Acc | $T$ Step | Acc |
| LIAF-Net (Wu et al., 2021)[TNNLS-2021] | ✗ | 10 | 70.4 | 60 | 97.6 |
| TA-SNN (Yao et al., 2021)[ICCV-2021] | ✗ | 10 | 72.0 | 60 | 98.6 |
| Rollout (Kugele et al., 2020)[Front. Neurosci-2020] | ✓ | 48 | 66.8 | 240 | 97.2 |
| DECOLLE (Kaiser et al., 2020)[Front. Neurosci-2020] | ✓ | - | - | 500 | 95.5 |
| tdBN (Zheng et al., 2021)[AAAI-2021] | ✓ | 10 | 67.8 | 40 | 96.9 |
| PLIF (Fang et al., 2021b)[ICCV-2021] | ✓ | 20 | 74.8 | 20 | 97.6 |
| SEW-ResNet (Fang et al., 2021a)[NeurIPS-2021] | ✓ | 16 | 74.4 | 16 | 97.9 |
| Dspike (Li et al., 2021)[NeurIPS-2021] | ✓ | 10 | 75.4* | - | - |
| SALT (Kim & Panda, 2021)[Neural Netw-2021] | ✓ | 20 | 67.1 | - | - |
| DSR (Meng et al., 2022)[CVPR-2022] | ✓ | 10 | 77.3* | - | - |
| **Spikformer** | ✓ | 10 | 78.9* | 10 | 96.9 |
| | ✓ | 16 | **80.9*** | 16 | **98.3** |

Spikformer-8-512 with 1 time step still achieves 70.14%. The above results show Spikformer is robust under low latency (fewer time steps) conditions.

**SSA** We conduct ablation studies on SSA to further identify its advantage. We first test its effect by replacing SSA with standard vanilla self-attention. We test two cases where Value is in floating point form (Spikformer-$L$-$D_w$ VSA V$_\mathcal{F}$) and in spike form (Spikformer-$L$-$D_w$ VSA).

We also test the different attention variants on ImageNet following Tab. 1. On CIFAR10, the performance of Spikformer with SSA is competitive compared to Spikformer-4-384$_w$ VSA and even Spikformer-4-384$_w$ VSA V$_\mathcal{F}$. On ImageNet, our Spikformer-8-512$_w$ SSA outperforms Spikformer-8-512$_w$ VSA by 0.68%. On CIFAR100 and ImageNet, the accuracy of Spikformer-$L$-$D_w$ VSA V$_\mathcal{F}$ is better than Spikformer because of the float-point-form Value. The reason why the Spikformer-8-512$_w$ I, Spikformer-8-512$_w$ ReLU, and Spikformer-8-512$_w$ LeakyReLU do

Table 5: Ablation study results on SSA, and time step.

| Datasets | Models | Time Step | Top1-Acc (%) |
|---|---|---|---|
| CIFAR10/100 | Spikformer-4-384$_w$ SSA | 1 | 93.51/74.36 |
| | | 2 | 93.59/76.28 |
| | | 4 | 95.19/77.86 |
| | | 6 | 95.34/78.61 |
| | Spikformer-4-384$_w$ VSA | 4 | 94.97/77.92 |
| | Spikformer-4-384$_w$ VSA V$_\mathcal{F}$ | 4 | 95.17/78.37 |
| ImageNet | Spikformer-8-512$_w$ I | 4 | ✗ |
| | Spikformer-8-512$_w$ ReLU | 4 | ✗ |
| | Spikformer-8-512$_w$ LeakyReLU | 4 | ✗ |
| | Spikformer-8-512$_w$ VSA | 4 | 72.70 |
| | Spikformer-8-512$_w$ VSA V$_\mathcal{F}$ | 4 | 73.96 |
| | Spikformer-8-512$_w$ SSA | 1 | 70.14 |
| | | 2 | 71.09 |
| | | 4 | 73.38 |
| | | 6 | 73.70 |

not converge is that the value of dot-product value of Query, Key, and Value is large, which makes the surrogate gradient of the output spike neuron layer disappear. More details are in the appendix D.4. In comparison, the dot-product value of the designed SSA is in a controllable range, which is determined by the sparse spike-form $Q$, $K$ and $V$, and makes Spikformer$_w$ SSA easy to converge.

## 5 CONCLUSION

In this work we explored the feasibility of implementing the self-attention mechanism and Transformer in Spiking Neuron Networks and propose Spikformer based on a new Spiking Self-Attention (SSA). Unlike the vanilla self-attention mechanism in ANNs, SSA is specifically designed for SNNs and spike data. We drop the complex operation of softmax in SSA, and instead perform matrix dot-product directly on spike-form Query, Key, and Value, which is efficient and avoids multiplications. In addition, this simple self-attention mechanism makes Spikformer work surprisingly well on both static and neuromorphic datasets. With directly training from scratch, Spiking Transformer outperforms the state-of-the-art SNNs models. We hope our investigations pave the way for further research on transformer-based SNNs models.

## REPRODUCIBILITY STATEMENT

Our codes are based on SpikingJelly(Fang et al., 2020), an open-source SNN framework, and Pytorch image models library (Timm)(Wightman, 2019). The experimental results in this paper are reproducible. We explain the details of model training and dataset augmentation in the main text and supplement it in the appendix. Our codes of Spikformer models are uploaded as supplementary material and will be available on GitHub after review.

## ACKNOWLEDGEMENTS

This work is supported by Nature Science Foundation of China (No.62202014 and No.62006007), Shenzhen Basic Research Program (No.JCYJ20220813151736001), and the National Innovation 2030 Major ST Project of China (No.2020AAA0104203).

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

APPENDIX

## A    MULTIHEAD SPIKING SELF ATTENTION

In practice, we reshape the $Q, K, V \in \mathbb{R}^{T \times N \times D}$ into multi-head form $\mathbb{R}^{T \times H \times N \times d}$, where $D = H \times d$. Then we split $Q, K, V$ into $H$ parts and run $H$ SSA operations, in parallel, which are called $H$-head SSA. The Multihead Spiking Self Attention (MSSA) is shown in follows:

$$Q = (q_1, q_2, \cdots, q_H), K = (k_1, k_2, \cdots, k_H), V = (v_1, v_2, \cdots, v_H) \quad q, k, v \in \mathbb{R}^{T \times N \times d} \quad (17)$$

$$\text{MSSA}^{'}(Q, K, V) = [\text{SSA}_1^{'}(q_1, k_1, v_1); \text{SSA}_2^{'}(q_2, k_2, v_2); \cdots; \text{SSA}_h^{'}(q_H, k_H, v_H)] \quad (18)$$

$$\text{MSSA}(Q, K, V) = \mathcal{SN}(\text{BN}(\text{Linear}(\text{MSSA}^{'}(Q, K, V)))) \quad (19)$$

## B    SPIKING SELF ATTENTION AND TIME STEP

In practice, $T$ is a independent dimension for spike neuron layer. In other layers, it is merged with the batch size.

## C    EXPERIMENT DETAILS

### C.1    TRAINING

Unlike the standard ViT, Dropout and Droppath are not applied in Spikformer. We remove the layer norm before each self-attention and MLP block, and add batch norm after each linear layer instead. In all Spikformer models, the hidden dimension of MLP blocks is $4 \times D$, where $D$ is the embedding dimension. As in Eq. (20), we select the Sigmoid function as the surrogate function with $\alpha = 4$.

$$\text{Sigmoid}(x) = \frac{1}{1 + \exp(-\alpha x)} \quad (20)$$

For DVS128 Gesture, we place a 1D max-pooling layer after $Q$ and $K$ to increase the density of the data, which improves the accuracy from 97.9% to 98.3% in 16 time steps. We set the threshold voltage $V_{th}$ of the spike neuron layer after $QK^{\mathrm{T}}V * s$ to 0.5, while the others are set to 1.

### C.2    THEORETICAL SYNAPTIC OPERATION AND ENERGY CONSUMPTION CALCULATION

The calculation of theoretical energy consumption requires first calculating the synaptic operations:

$$\text{SOPs}(l) = fr \times T \times \text{FLOPs}(l) \quad (21)$$

where $l$ is a block/layer in Spikformer, $fr$ is the firing rate of the input spike train of the block/layer and $T$ is the simulation time step of spike neuron. $\text{FLOPs}(l)$ refers to floating point operations of $l$, which is the number of multiply-and-accumulate (MAC) operations. And $\text{SOPs}$ is the number of spike-based accumulate (AC) operations. We estimate the theoretical energy consumption of Spikformer according to (Kundu et al., 2021b; Hu et al., 2021b; Horowitz, 2014; Kundu et al., 2021a; Yin et al., 2021; Panda et al., 2020; Yao et al., 2022). We assume that the MAC and AC operations are implemented on the 45nm hardware [12], where $E_{MAC} = 4.6pJ$ and $E_{AC} = 0.9pJ$. The theoretical energy consumption of Spikformer is calculated:

$$E_{Spikformer} = E_{MAC} \times \text{FL}^1_{\text{SNN Conv}}$$
$$+ E_{AC} \times \left( \sum_{n=2}^{N} \text{SOP}^n_{\text{SNN Conv}} + \sum_{m=1}^{M} \text{SOP}^m_{\text{SNN FC}} + \sum_{l=1}^{L} \text{SOP}^l_{\text{SSA}} \right) \quad (22)$$

where $\mathrm{FL}^1_{SNN\ Conv}$ is the first layer to encode static RGB images into spike-form. Then the SOPs of $m$ SNN Conv layers, $n$ SNN Fully Connected Layer (FC) and $l$ SSA are added together and multiplied by $E_{AC}$. For ANNs, the theoretical energy consumption of block $b$ is calculated:

$$\mathrm{Power}(b) = 4.6pJ \times \mathrm{FLOPs}(b) \tag{23}$$

For SNNs, $\mathrm{Power}(b)$ is:

$$\mathrm{Power}(b) = 0.9pJ \times \mathrm{SOPs}(b) \tag{24}$$

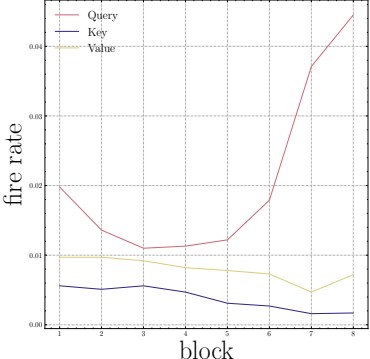

fire rate of $Q, K, V$

Figure 4: Fire rate of Query, Key and Value of blocks in Spikformer-8-512 on ImageNet test set.

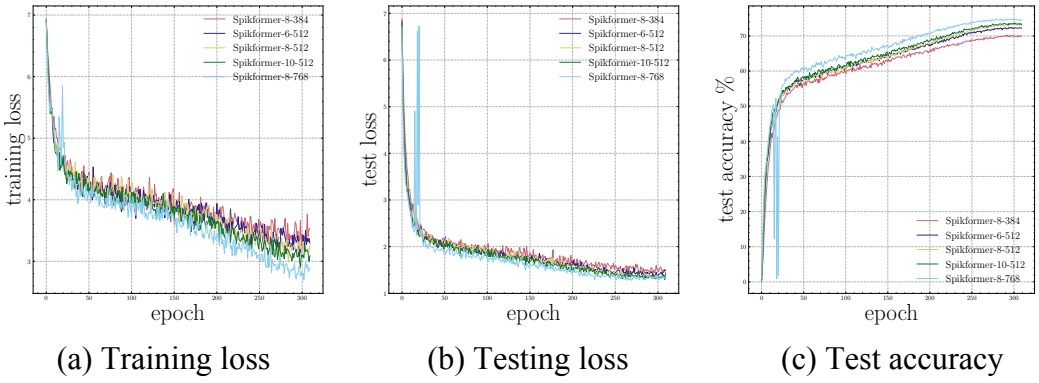

(a) Training loss       (b) Testing loss       (c) Test accuracy

Figure 5: Training loss, testing loss and test accuracy on ImageNet.

# D    ADDITIONAL RESULTS

## D.1    FIRE RATE OF QUERY, KEY AND VALUE

As shown in 4, the Query, Key and Value are very spare in SSA, causing sparse computation of SSA.

## D.2    LOSS AND ACCURACY ON IMAGENET

We show the training loss, testing loss and test accuracy of Spikformer in Figue. 5. Both training and testing losses decrease as the number of Spikformer blocks increases or the embedding dimension increases.

Table 6: Additional result on CIFAR10/100. Spikformer-4-384$_{w \text{ IF}}$ uses the Integrate-and-Fire neuron.

| Models | Time Step | Top1-Acc (%) |
|---|---|---|
| Spikformer-4-384$_{w \text{ I}}$ | 1 | 92.39/74.28 |
| Spikformer-4-384$_{w \text{ ReLU}}$ | 1 | 92.98/74.32 |
| Spikformer-4-384$_{w \text{ LeakyReLU}}$ | 1 | 92.88/74.31 |
| Spikformer-4-384$_{w \text{ VSA}}$ | 1 | 93.11/74.37 |
| Spikformer-4-384$_{w \text{ IF}}$ | 4 | 95.33/78.14 |

Table 7: Transfer Learning on CIFAR10/100.

| Models | CIFAR10 | CIFAR100 |
|---|---|---|
| Spikformer-4-384 | 95.54 | 79.96 |
| Spikformer-8-384 | 96.64 | 82.09 |
| Spikformer-8-512 | 97.03 | 83.83 |

### D.3 Additional Accuracy Results on CIFAR

We conduct additional experiments on CIFAR as shown in Tab. 6.

### D.4 Analysis of self-attention variants not converging on ImageNet

The reason that the three models do not converge in Tab. 5 is explain as follows. As shown in Figure. 6 (a), the gradient of sigmoid surrogate function vanishes when the difference between the average input value $V_i$ and the firing threshold $V_{th}$ is too large or too small. We collect the output value of $QK^{\mathrm{T}}V * s$ after one training eopch of Spikformer-8-512$_{w \text{ I}}$, Spikformer-8-512$_{w \text{ ReLU}}$, Spikformer-8-512$_{w \text{ LeakyReLU}}$, and Spikformer-8-512$_{w \text{ SSA}}$, which will be sent to the spike neuron layer as the input value $V_i$, as shown in Eq. (15). Compared to the other three variants, as shown in Figure. 6 (b), the value of $QK^{\mathrm{T}}V * s$ in Spikformer-8-512$_{w \text{ SSA}}$ is controlled in a suitable range. Therefore, SSA has stable surrogate gradients during training and converges easily.

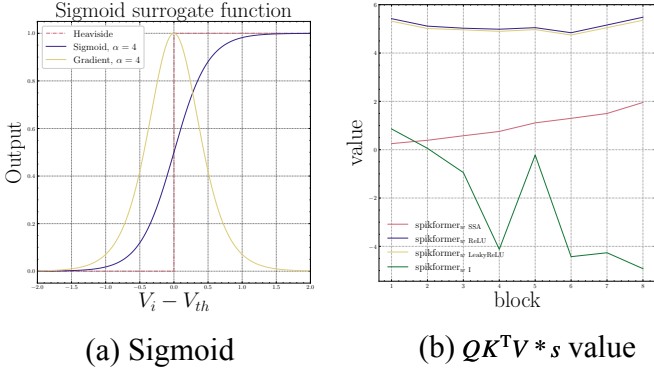

(a) Sigmoid  (b) $QK^{\mathrm{T}}V * s$ value

Figure 6: (a) the sigmoid surrogate function and its gradient curve. (b) the value of $QK^{\mathrm{T}}V$.

### D.5 Transfer Learning

We transfer Spikformer to the downstream CIFAR dataset. The pre-trained Spikformer-4-384 and Spikformer-8-384/512 on ImageNet are finetuned with 60 epochs. The input size of CIFAR is $224 \times 224$. The remaining hyperparameters are the same as the ones directly trained on CIFAR. As shown in Tab. 7, Spikformer shows high transfer ability.

