# OpenReview forum: "Spikformer: When Spiking Neural Network Meets Transformer "
_ICLR.cc/2023/Conference — ICLR 2023 poster_

### Official Review · Reviewer_cw3J · 2022-10-22

**Confidence:** 5
**Correctness:** 3
**Technical Novelty And Significance:** 3
**Empirical Novelty And Significance:** 3
**Recommendation:** 8

**Clarity, Quality, Novelty And Reproducibility:**

The motivation is straightforward and the paper is overall clearly written. However, this article is slightly less innovative and less rigorous in spite of the superior results.

**Strength And Weaknesses:**

Strength:
1.	The first time to implement self-attention and transformer in large-scale model and dataset.
2.	The novel module pays attention to spike-based operations and avoids multiplications to obtain high energy efficiency.
3.	The model achieves new state-of-the-art results on various datasets.

Weakness:
1.	The paper provides much about the experimental results, while the contents for the method and motivation seem insufficient.
2.	Not much insights have been given as to why spiking self-attention should be designed in this way.
3.	Other concerns detailed in Summary.

**Summary Of The Paper:**

In this paper, the authors provide a feasible implementation of SNN-oriented self-attention mechanism and Vision Transformer, obtaining superior results on both static and neuromorphic benchmarks.

**Summary Of The Review:**

1. In Eqn. (16), additional SN-BN-Linear is applied to SSA’. What is the role of this extra part?
2. Table 1. ①Why is the ratio of (SOPs of A_SSA)/(FLOPs of A_ReLU) much smaller than those on other datasets? ② It is stated in the paper that the VSA with float-point-form Q, K and softmax is redundant for spike-form X, V, which cannot get more information from X, V than SSA. Does it mean that a continuous attention map cannot work better than the spike-formed attention map? Can the authors provide more explanations for this and why the opposite conclusion is drawn from Table 5? ③ It seems that sparsity has been taken into account for both SOPs and FLOPs (in ReLU), but this is not explicitly stated and seems confusing.
3. The authors use the data 77fJ/SOP for their energy estimation. However, it only stands for the Energy per Synaptic Event reported in ROLLS, while there are other operations like producing a spike which consumes 3.7pJ. The authors should provide a more rigorous comparison in terms of energy consumption.
4. I noticed that the model used skip connections, and I am curious about how the two branches of spikes are merged. If the two spike trains are added together directly, the data will not be limited to the pure spike form and then does the first linear layer in the subsequent block still avoid multiplications?
5. What is the scaling factor s in Eqn. (15)?
6. In Table 3, it would be better to compare with ANN-Transformers instead of ANN-ResNet19.

---

> ### Author Response · Authors · 2022-11-11
> **Response to Reviewer cw3J (1/3)**
>
> Thank you for your thoughtful comments, which prompt us to improve our manuscript. We would like to address your concerns and answer your questions in the following.
> >Weakness:
> >1. The paper provides much about the experimental results,  while the contents for the method and motivation seem insufficient.
>
> **R:** The most serious problem with current SNNs is their poor performance. There is a large performance gap between it, and ANN. **ANN-Transformer has a higher ceiling than ANN-ConvNet when training with a large amount of data [1, 2, 3]. Introducing the Transformer architecture to SNN is of great significance, and Spikformer, as the first SNN-Transformer, can enrich architecture and improve performance of SNN.**  Transformers have developed rapidly and have many advanced methods in ANN. **We hope our Spikformer can shed light on transformer-based research in SNN.** Based on Spikformer, we can conduct many studies to use the advantages of Transformer and SNN further. **For example, a spike sequence can be regarded as a natural mask. Currently, we are also exploring Spikformer-based self-supervised pretraining to improve the performance of SNN.**
> **Overall, our motivation to introduce Transformer to SNN is simple but very strong.**
>
> >2. Not much insights have been given as to why spiking self-attention should be designed in this way.
>
> **R:** **There are four aspects to the design of our spiking self-attention (SSA):**
>
> **①** We think that the Query, Key, and Value **should be in spike form**, not only following the computational feature of SNN but also probably being easier to implement on hardware than in floating-point form.
>
> **②** The obstacles to applying self-attention in SNNs are mainly caused by softmax. However, the most crucial role of Softmax in vanilla self-attention is keeping the attention map non-negative [4]. Spike-form Query and Key are naturally non-negative, leading a non-negative attention map. That is why we choose to discard the softmax function.
>
> **③** We have tried other designs and empirically found the current SSA suitable design. For example, we set one of the Query, Key, and Value to the floating-point number, and the remaining are spike forms, which can also avoid multiplication in self-attention. Nevertheless, it does not get a better effect than SSA. The reason may be that the large value caused by matrix multiplication $QK^TV$ will disappear the gradient of the surrogate function, which is similar to that explained in Appendix D.4 of the manuscript.
>
> **④** Considering of above points, **sparse spike-form $Q,K,V$ allow SSA to discard softmax. That is, the data chrematistic in SNN solves the obstacle to self-attention implementation in SNN, which causes the first SNN-self-attention.**
>
>
> >1. In Eqn. (16), additional SN-BN-Linear is applied to SSA’.  What is the role of this extra part?
>
> **R:** **The SN-BN-Linear in SSA is used to transform and fuse the channel information,  especially after the concatenation of the multi-head feature,  which is a common structure in the self-attention block [5, 6].**  The reviewer's comments let us discover a mistake in the paper. $\rm{MSSA}(Q,K,V)$ should concatenate the ${\rm{SSA}}^{'}(Q,K,V)$ and contain SN-BN-Linear after concatenation. Therefore, we modify Eqn. 18 as follows.
> $$
> {\rm{MSSA}}^{'}(Q,K,V) = [ {\rm{SSA}}^{'}_1(q_1,k_1,v_1); {\rm{SSA}}^{'}_2(q_2,k_2,v_2); \cdots ; {\rm{SSA}}^{'}_h(q_H,k_H,v_H)]
> $$
> **Eq. 18 in the Paper**
> $$
>     {\rm{MSSA}}(Q,K,V)={\mathcal{SN}}({\rm{BN}}({\rm{Linear}}({\rm{MSSA}}^{'}(Q,K,V))))
> $$
> **Eq. 19 in the Paper**
>
> 2.
> >① Why is the ratio of (SOPs of A_SSA)/(FLOPs of A_ReLU) much smaller than those on other datasets?
>
> **R:** The low ratio of SOPs of $A_{\rm SSA}$ is because the $Q,K,V$ involved in calculation is all in sparse spike form. Taking Spikformer-8-512 as an example, as shown in Fig. 4 in appendix D.1, the fire rate of $Q, K, V$ in each block is very low (less than 0.05).
> 	We miscalculated the FLOPs of $A_{\rm ReLU}$, which should be equal to the FLOPs of $A_{\rm I}$ and $A_{\rm LeakyReLU}$. We modify it in Tab. 1.

---

> > ### Author Response · Authors · 2022-11-11
> > **Response to Reviewer cw3J (2/3)**
> >
> > >② It is stated in the paper that the VSA with float-point-form Q, K and softmax is redundant for spike-form X, V, which cannot get more information from X, V than SSA. Does it mean that a continuous attention map cannot work better than the spike-formed attention map? Can the authors provide more explanations for this and why the opposite conclusion is drawn from Table 5?
> >
> > **R: In the experiments, VSA is divided into two types according to the form of Value:**
> >
> > ① $\pmb{\rm VSA_{VS}}$ with spike-form Value, which is our **default** $\pmb{\rm VSA}$
> >
> > ② $\pmb{\rm VSA_{VF}}$ with floating-point form Value.
> >
> > We show the performance of Spikformer with $\rm SSA$, $\rm VSA_{VS}$ and ${\rm VSA_{VF}}$ on CIFAR and ImageNet in Tab. R4-1 and R4-2, which is from Tab. 5 of the manuscript.
> > As shown bellow, $\pmb{\rm SSA}$ is competitive with $\rm VSA_{VS}$ ($\pmb{\rm VSA}$), and does not perform as well as ${\rm VSA_{VF}}$.
> >
> > **So we do not get the opposite conclusion in Tab. 5.**
> >
> > | Spikformer-4-384 | CIFAR10 Acc (%)|CIFAR100 Acc (%)|
> > |:-----------------:|:--------:|:---------------:|
> > |$\rm SSA$ |  **95.19**    | 77.86  |
> > |$\rm VSA_{VS}$ |  94.97    |  **77.92** |
> > |${\rm VSA_{VF}}$ |   95.17   | **78.37**  |
> >
> > **Table R4-1: Comparison of $\rm SSA$, $\rm VSA_{VS}$ and ${\rm VSA_{VF}}$ on CIFAR .**
> >
> > Spikformer-8-512  |ImageNet Acc (%) |
> > |-----|:-----:|
> > |$\rm SSA$ |  73.38    |
> > |$\rm VSA_{VS}$ |  72.70    |
> > |${\rm VSA_{VF}}$ |   **73.96**    |
> > **Table R4-1: Comparison of $\rm SSA$, $\rm VSA_{VS}$ and ${\rm VSA_{VF}}$ on ImageNet .**
> >
> > ① When the Value is floating-point form, ${\rm VSA_{VF}}$ works better than $\rm SSA$. **But ${\rm{VSA}}_{\rm V{{F}}}$ does not fit SNN's computational characteristics at all.**
> >
> > ②  When processing spike-form $X,V$, $\rm VSA_{VS}$ (${\rm VSA}$) is competitive with  our $\rm SSA$. **But $\rm VSA_{VS}$  also can not meet the computing characteristics of SNN.**  **In comparison, our $\rm SSA$ not only maintains competitive performance but also conforms to the efficient computation of SNN.**
> > In other words, when handling spike-form $X, V$ in SNN, ${\rm VSA}$, with floating-point form $Q, K$, and softmax, achieves similar accuracy compared to $\rm SSA$, with spike-form $Q, K$ and discarding softmax.  In addition, $\rm SSA$ computes more efficiently than ${\rm VSA}$.
> >
> > Therefore,  the floating-point-form $Q, K$ and softmax, which belongs to $\pmb{\rm VSA}$ ($\rm VSA_{VS}$),  is redundant for processing spike-form $X, V$.
> > Based on the above observations, we conclude that $\pmb{\rm SSA}$ is more suitable for SNN than $\pmb{\rm VSA}$.
> >
> > >③ It seems that sparsity has been taken into account for both SOPs and FLOPs (in ReLU), but this is not explicitly stated and seems confusing.
> >
> > **R:** Yes,  it is wrong to take the sparsity into account for FLOPs (In ReLU) and we have made corresponding modifications in Tab. 1. Thanks for your correction.
> >
> > >3. The authors use the data 77fJ/SOP for their energy estimation. However, it only stands for the Energy per Synaptic Event reported in ROLLS, while there are other operations like producing a spike which consumes 3.7pJ. The authors should provide a more rigorous comparison in terms of energy consumption.
> >
> > **R:** We initially calculated theoretical energy consumption based on the methods of the papers [7] and [8], both of which are based on data from ROLLS [9]. After the reviewer's a kind reminder, we re-estimate the theoretical energy consumption of Spikformer according to the paper [10, 11, 12, 13, 14, 15, 16], in which $E_{\rm MAC}=4.6pJ$ and $E_{\rm AC}=0.9pJ$. For static images, the energy estimation of Spikformer  $ E_{\rm Spikformer}$ is shown below.
> >
> > $$
> > E_{\rm Spikformer} = E_{\rm MAC}  \times {\rm FL}^1_{\rm SNNConv} + E_{\rm AC} \times 		  \left(\sum_{n=2}^{N}{\rm SOP}^n_{{\rm SNNConv}} + \sum_{m=1}^{M}{\rm SOP}^m_{{\rm SNNFC}} + \sum_{l=1}^L{\rm SOP}^l_{{\rm SSA}}\right)
> > $$
> >
> > where ${\rm FL}^1_{\rm SNNConv}$ is the first layer to encode static RGB images into spike form. Then the SOPs of $m$  SNN Conv,  $n$ SNN Fully Connected Layer (FC), and $l$ SSA are added together and multiplied by $E_{\rm AC}$. For the element-wise addition of spike sequences between Spikformer blocks, we also use $E_{\rm AC}=0.9pJ$, as  $0.9pJ$ is 32-bit floating-point addition which can be regarded as containing binary spike addition and multi-bit spike addition [13]. In addition, multi-bit spike processing represents a negligible extra energy cost compared to binary spike processing which is stated in the file of Loihi 2 [17]. We modify the value of theoretical energy consumption in Fig. 1, Tab.1, and Tab.2, and update the computation method in App. C.2.

---

> > > ### Author Response · Authors · 2022-11-11
> > > **Response to Reviewer cw3J (3/3)**
> > >
> > > >4. I noticed that the model used skip connections, and I am curious about how the two branches of spikes are merged. If the two spike trains are added together directly, the data will not be limited to the pure spike form and then does the first linear layer in the subsequent block still avoid multiplications?
> > >
> > > **R:** Two branches of spikes are added in an element-wise way (added directly) following SEW ResNet [8], which is a relatively popular SNN residual architecture[18, 19, 20]. After addition operation, the input to the subsequent block is indeed no longer binary spikes but multi-bit spikes.  Convolution layer and Linear layer are linear operations. Therefore, computations in which multi-bit spikes are fed into the layer can be decomposed into addition operations and multiplication is avoided, which is shown as
> > >
> > > $$
> > > {\rm Linear}(X_1 + X_2 + \cdots +X_{n}) =  {\rm Linear}(X_1) + {\rm Linear}(X_2) + \cdots + {\rm Linear}(X_n)
> > > $$
> > > $$
> > > {\rm Conv}(X_1 + X_2 + \cdots +X_{n}) =  {\rm Conv}(X_1) + {\rm Conv}(X_2) + \cdots + {\rm Conv}(X_n)
> > > $$
> > > where $X_1,X_2,\cdots X_{n-1}$ is the skip connections of  binary spikes and $X_n$ is the current  binary spikes input.  Furthermore,  some neuromorphic chips can handle multi-bit spikes. For example, the Tianjic chip [21] supports 8-bit spike gradation, and Loihi can process up to 32-bit spike payload [17].
> > >
> > > > 5. What is the scaling factor s in Eqn. (15)?
> > >
> > > **R:** **The scaling factor is used to control the large value of the matrix multiplication result of $Q, K$, which is originally used in ANN-transformer [1, 2] and is $\sqrt d$ shown in Eq. 12 of the manuscript. It does not affect the avoidance of multiplications of SSA, because $Q,K,V$ are spike forms, which can be successively conducted addition on it.**  Although matrix multiplication between $Q, K$, and $V$ involves only sparse addition operations, the results of $QK^T$ may still yield large values. For example,  when $Q, K \in 64\times 32$, the largest value of $QK^T$ is 32, which may be even bigger after multiplying $V$. The above large value will cause the surrogate gradient of the spiking neuron to vanish during training. The use of a scaling factor is conducive to model convergence. **We also have trained the Spikformer-4-384 without a scaling factor on CIFAR10 and only got $68.44\%$ accuracy, so the scaling factor is also an important value of our spikformer.**
> > >
> > > >6. In Table 3, it would be better to compare with ANN-Transformers instead of ANN-ResNet19.
> > >
> > > **R:** Thanks for your suggestion. **We have added a performance comparison between Spikformer and ANN-Transformer in Tab. 1, Tab. 3 and analysed it in Sec.4.1.**
> > >
> > > **On ImageNet**, ANN-Transformer-8-512 is 7.42% higher than Spikformer-8-512 (**80.80% VS. 73.38%**), but the theoretical energy consumption is 3.31× of Spikformer-8-512 (**38.34 mJ VS. 11.58 mJ**).
> > >
> > > **On CIFAR10/100**, Spikformer-4-384 achieves 95.19% and 77.86% and ANN-Transformer-4-384 achieves 96.73% and 81.02% after 300 epoch training.
> > > At present ANN-transformer is better than our Spikformer, which is normal, but in the future, we will strive to narrow the performance gap.

---

> > > > ### Author Response · Authors · 2022-11-11
> > > > **Reference**
> > > >
> > > > [1] d’Ascoli, Stéphane, et al. "Convit: Improving vision transformers with soft convolutional inductive biases." _International Conference on Machine Learning_. PMLR, 2021.
> > > >
> > > > [2] Wang, Cong, et al. "Convolutional Embedding Makes Hierarchical Vision Transformer Stronger." _European Conference on Computer Vision. Springer_, Cham, 2022.
> > > >
> > > > [3] Chen, Zhengsu, et al. "Visformer: The vision-friendly transformer."  _In Proceedings of the IEEE/CVF International Conference on Computer Vision_. 2021.
> > > >
> > > > [4] Zhen Qin, Weixuan Sun, Hui Deng, Dongxu Li, Yunshen Wei, Baohong Lv, Junjie Yan, Lingpeng Kong, and Yiran Zhong. cosformer: _Rethinking softmax in attention. In International Conference on Learning Representations (ICLR)_, 2022.
> > > >
> > > > [5] Vaswani, A., Shazeer, N., Parmar, N., Uszkoreit, J., Jones, L., Gomez, A. N., ... & Polosukhin, I. (2017). Attention is all you need.  _Advances in neural information processing systems (NeurIPS)_, 30.
> > > >
> > > > [6] Dosovitskiy, A., Beyer, L., Kolesnikov, A., Weissenborn, D., Zhai, X., Unterthiner, T., ... & Houlsby, N. (2020). An image is worth 16x16 words: Transformers for image recognition at scale. _In International Conference on Learning Representations (ICLR)_, 2020.
> > > >
> > > > [7] Hu, Y., Tang, H., & Pan, G. (2018). Spiking Deep Residual Networks. _IEEE Transactions on Neural Networks and Learning Systems_.
> > > >
> > > > [8] Fang, W., Yu, Z., Chen, Y., Huang, T., Masquelier, T., & Tian, Y. (2021). Deep residual learning in spiking neural networks. _Advances in Neural Information Processing Systems (NeurIPS)_, 34, 21056-21069.
> > > >
> > > > [9] Indiveri, G., Corradi, F., & Qiao, N. (2015, December). Neuromorphic architectures for spiking deep neural networks. In _2015 IEEE International Electron Devices Meeting (IEDM)_ (pp. 4-2). IEEE.
> > > >
> > > > [10] Kundu, S., Pedram, M., & Beerel, P. A. (2021). Hire-snn: Harnessing the inherent robustness of energy-efficient deep spiking neural networks by training with crafted input noise. _In Proceedings of the IEEE/CVF International Conference on Computer Vision (pp. 5209-5218)_.
> > > >
> > > > [11] Hu, Y., Wu, Y., Deng, L., & Li, G. (2021). Advancing Residual Learning towards Powerful Deep Spiking Neural Networks. _arXiv preprint arXiv:2112.08954_.
> > > >
> > > > [12] Yao, M., Zhao, G., Zhang, H., Hu, Y., Deng, L., Tian, Y., ... & Li, G. (2022). Attention Spiking Neural Networks. _arXiv preprint arXiv:2209.13929_.
> > > >
> > > > [13] Mark Horowitz. 1.1 computing’s energy problem (and what we can do about it). _In 2014 IEEE International Solid-State Circuits Conference Digest of Technical Papers (ISSCC)_, pages 10–14, 2014.
> > > >
> > > > [14] Kundu, S., Datta, G., Pedram, M., & Beerel, P. A. (2021). Spike-thrift: Towards energy-efficient deep spiking neural networks by limiting spiking activity via attention-guided compression. _In Proceedings of the IEEE/CVF Winter Conference on Applications of Computer Vision_ (pp. 3953-3962).
> > > >
> > > > [15] Yin, B., Corradi, F., & Bohté, S. M. (2021). Accurate and efficient time-domain classification with adaptive spiking recurrent neural networks. Nature Machine Intelligence, 3(10), 905-913.
> > > >
> > > > [16]Panda, P., Aketi, S. A., & Roy, K. (2020). Toward scalable, efficient, and accurate deep spiking neural networks with backward residual connections, stochastic softmax, and hybridization. _Frontiers in Neuroscience_, 14, 653.
> > > >
> > > > [17] Taking Neuromorphic Computing to the Next Level with Loihi 2. https://download.intel.com/newsroom/2021/new-technologies/neuromorphic-computing-loihi-2-brief.pdf
> > > >
> > > > [18] Hagenaars, J., Paredes-Vallés, F., & De Croon, G. (2021). Self-supervised learning of event-based optical flow with spiking neural networks. _Advances in Neural Information Processing Systems_, 34, 7167-7179.
> > > >
> > > > [19] Chen, Yanqi, et al. "State Transition of Dendritic Spines Improves Learning of Sparse Spiking Neural Networks." _International Conference on Machine Learning_. PMLR, 2022.
> > > >
> > > > [20] Vicente-Sola, A., Manna, D. L., Kirkland, P., Di Caterina, G., & Bihl, T. (2022). Keys to accurate feature extraction using residual spiking neural networks. _Neuromorphic Computing and Engineering_, 2(4), 044001.
> > > >
> > > > [21] Pei, Jing, et al. "Towards artificial general intelligence with hybrid Tianjic chip architecture." Nature 572.7767 (2019): 106-111.
> > > >
> > > > [22] Hassani, Ali, et al. "Escaping the big data paradigm with compact transformers." arXiv preprint arXiv:2104.05704 (2021).

---

> ### Author Response · Authors · 2022-11-17
> **Discussion Period Ending Soon**
>
> Dear reviewer cw3J,
>
> Thank you again for your valuable time and insightful comments. We have provided thorough responses to you and sincerely hope you can look through them and update the scores if your concerns have been resolved. We are also open to further discussion if the concerns have not been fully addressed. Please feel free to let us know if you still have any questions.
>
> Best regards!
>
> Authors

---

> > ### Comment · Reviewer_cw3J · 2022-11-22
> > **Response to rebuttal**
> >
> > Thanks for the authors' kind responses that can address my concerns.

---

### Official Review · Reviewer_vg9k · 2022-10-24

**Confidence:** 4
**Correctness:** 4
**Technical Novelty And Significance:** 4
**Empirical Novelty And Significance:** 4
**Recommendation:** 10

**Clarity, Quality, Novelty And Reproducibility:**

Clarity
The paper is very clearly written. The different aspects of the proposed architecture are clearly outlined both in the text and the figures of the paper. The paper clearly highlights its novel contributions and makes references to key relevant related work.

Quality
The quality of the paper is sound. The claims in the paper are backed by a sufficient number of relevant experiments comparing Spikformers with similar SNN architectures. The theoretical parts of paper are solid and I cannot find any obvious errors in them.

Novelty
The authors introduce a novel version of self-attention for SNNs. This is the first attempt to implement self-attention in SNNs that I am aware of.

Reproducibility
The authors provide enough information to enable others to reproduce their work. I have not personally attempted to reproduce the key results of the paper, but I believe this will be easily possible once the source code is released.

**Strength And Weaknesses:**

Strengths
The paper's motivation is a relatively simple idea (implement a SNN version of Transformers), but its implementation is non-trivial given the constraints of SNNs. The authors' solution is a simple and efficient one, which sets solid ground for future iterations. The paper strengths are the novel architecture contributions and the SOTA results when compared with other SNNs.

Weaknesses
Even though it is not the authors' motivation, it can be argued that this paper does not offer an alternative architecture to Transformers. Transformers likely outperform Spikformers in many ways and most importantly in prediction quality. On the other hand, we are also not learning about new biological features that are important for brain computing. This paper is about advancing the field of SNN research, which has not yet produced an algorithm that can broadly compete with ANNs. Yes, we know for a fact that NNs in the brain are SNNs, but we still do not know why, and this paper is obviously not helping us clarify that.

**Summary Of The Paper:**

The authors introduce Spikformer a SNN version of Transformer, which includes a novel implementation of self-attention with spiking representations. The authors show that their proposed architecture outperforms SOTA SNNs on both traditional image tasks as well as neuromorphic tasks.

**Summary Of The Review:**

Overall a solid contribution to the field of SNNs research and one that should deserve a spot in the conference.

---

> ### Author Response · Authors · 2022-11-12
> **Response to Reviewer vg9k**
>
> Thank you for your recognition of our work. And now we will explain and discuss the weaknesses you raised.
>
> > Even though it is not the authors' motivation, it can be argued that this paper does not offer an alternative architecture to Transformers. Transformers likely outperform Spikformers in many ways and most importantly in prediction quality.
>
> **R:** Indeed, Transformer is currently performing better than Spikformer. Mainly because the discrete spikes in the SNN would lose information.
> As the advantage of SNN is in the power consumption, we show the performance and theoretical energy consumption of different models trained on ImageNet. ANN-Transformer-8-512 is 7.42% higher than Spikformer. Spikformer has much lower theoretical energy consumption (**11.58mJ VS. 38.34mJ**).
> **We will study more advanced and effective Spikformer variants in future work to narrow the gap with Transformer.**
>
> | Methods| Power(mJ)|Accuracy(\%)|
> |-------------|:--------:|:---------------:|
> |ANN-ResNet-152 |  53.36   |78.30   |
> |SNN-ResNet-152 |  12.89   |69.26   |
> |ANN-Transformer-8-512 | 38.34    |**80.80**  |
> |Spikformer-8-512 | **11.58**    |73.38  |
> **Table R3-1: Comparison of power consumption.**
>
> > On the other hand, we are also not learning about new biological features that are important for brain computing. This paper is about advancing the field of SNN research, which has not yet produced an algorithm that can broadly compete with ANNs. Yes, we know for a fact that NNs in the brain are SNNs, but we still do not know why, and this paper is obviously not helping us clarify that.
>
> **R:** The study of new biological features requires computer science and biological science joint efforts. At present, there is a large performance gap between ANN and SNN. We hope to continue to narrow the gap with ANN in the future and combine brain science and SNN with exploring new biological characteristics and relevant rationality explanations.

---

### Official Review · Reviewer_6yty · 2022-10-25

**Confidence:** 5
**Correctness:** 2
**Technical Novelty And Significance:** 2
**Empirical Novelty And Significance:** Not applicable
**Recommendation:** 3

**Clarity, Quality, Novelty And Reproducibility:**

The novelty of this paper is limited, and the experimental evaluation is flawed and fails to adequately support the main claims.

**Strength And Weaknesses:**

++ The figures are nice

--The novelty of this paper is limited. The major contribution is that the authors apply the transformer framework to SNNs. The performance of the proposed spiking transformer is lower than the transformer of ANNs. This research cannot give full play to SNN strengths, like temporal processing capability, and will cause researchers to go astray.

--The authors over-claimed their contributions. The authors claim that they are the first to implement Transformer in SNNs. Actually, there exist many spiking transformer works, e.g., [1-3].

-- The authors use floating point operations to add spikes, which defeats the purpose of SNNs.

--The authors are not familiar with the research progress of SNNs. There are many of the latest works, which are ignored by the authors. E.g. [4-7].

--The comparison is unfair. As the network structure of the spiking transformer differs from SEW-ResNet-152, there is no doubt the performance will be better. I do not think it is a fair comparison. Please show the performance of the corresponding ANN.

--Ignoring all that, the results are not convincing. The performance of the proposed method is not SOTA. The authors only chose favourable comparisons. For example, TET achieves 83.17% accuracy on CIFAR10-DVS dataset with 10 steps, while the proposed method gets 80.9% with 16 time-steps.

--The authors did not compare the training cost of the methods. Please add it in Table 2.

--The ANN2SNN method achieves better performance with the increase of time-steps. How about spiking transformer? If you train the network with time-steps 4 on ImageNet, please show the performance of the network when the time-steps are 2,3,10,20,50.

--Please compare SOPs and Power in Table 3 and 4.

[1] Zhang, Jiqing, et al. "Spiking Transformers for Event-Based Single Object Tracking." CVPR. 2022.

[2] Zhang, Jiyuan, et al. "Spike Transformer: Monocular Depth Estimation for Spiking Camera."ECCV. 2022

[3] Mueller, Etienne, et al. "Spiking Transformer Networks: A Rate Coded Approach for Processing Sequential Data." 2021 7th International Conference on Systems and Informatics (ICSAI). IEEE, 2021.

[4] Xiao, Mingqing, et al. "Training feedback spiking neural networks by implicit differentiation on the equilibrium state." Advances in Neural Information Processing Systems 34 (2021): 14516-14528.

[5] Meng, Qingyan, et al. "Training much deeper spiking neural networks with a small number of time-steps." Neural Networks153 (2022): 254-268.

[6] Bu, Tong, et al. "Optimal ANN-SNN Conversion for High-accuracy and Ultra-low-latency Spiking Neural Networks." International Conference on Learning Representations. 2022.

[7] Wang, Yuchen, et al. "Signed Neuron with Memory: Towards Simple, Accurate and High-Efficient ANN-SNN Conversion." International Joint Conference on Artificial Intelligence. 2022.

**Summary Of The Paper:**

This paper proposes the spiking version of the transformer by designing spike-form of Query, Key and Value of self-attention. The experiments are performed on both static and neuromorphic datasets for the classification tasks.

**Summary Of The Review:**

Overall, the significance of this work is limited, and the authors over-claimed their contributions.

---

> ### Author Response · Authors · 2022-11-11
> **Response to Reviewer 6yty (1/3)**
>
> We appreciate your detailed comments. We would like to address your concerns below.
>
> > The novelty of this paper is limited. The major contribution is that the authors apply the transformer framework to SNNs.
>
> **R:**  **In our paper, Spikformer provides a new alternative backbone for SNN, which first introduces the Transformer architecture to SNN.** Especially, Transformer has been widely proven with higher model capacity and great potential than ConvNet, while it has yet to be applied to the SNN backbone before our proposed Spikformer. Besides, applying the Transformer to the SNN backbone is not straightforward. We have made a great effort to make full use of the characteristics of both SNN and Transformer in Spikformer, and argue that introducing the Transformer architecture to SNN is of great significance. Spikformer, as the first SNN-Transformer backbone, can enrich SNN's architecture and improve SNN's performance.
>
> >The performance of the proposed spiking transformer is lower than the transformer of ANNs.
>
> **R:**   **SNN has the superiority of lower power consumption than ANN on specific hardware [1], though the performance is worse than advanced ANN because discrete spikes would lose some information.**  Therefore, it is normal for Spikformer to have lower performance than ANN-Transformer. **The case is similar to SNN (SEW) ResNet [2]. For example, ANN-ResNet-152 is 9.04% higher than SNN-ResNet-152.** However, we narrow down the performance gap between advanced ANN and directly trained SNN and improve the SOTA results of directly trained SNN.
> As the advantage of SNN is in the power consumption, we show the performance and theoretical energy consumption of different models trained on ImageNet. ANN-Transformer-8-512 is 7.42% higher than Spikformer, Spikformer has much lower theoretical energy consumption (**11.58mJ VS. 38.34mJ**).
>
> | Methods| Power(mJ)|Accuracy(\%)|
> |-------------|:--------:|:---------------:|
> |ANN-ResNet-152 |  53.36   |78.30   |
> |SNN-ResNet-152 |  12.89   |69.26   |
> |ANN-Transformer-8-512 | 38.34    |**80.80**  |
> |Spikformer-8-512 | **11.58**    |73.38  |
> **Table R2-1: Comparison of power consumption.**
>
> The most serious problem with current SNNs is their poor performance. **We hope that Spikformer serving as a backbone can enhance the performance of SNN and narrow the performance gap between ANN and directly trained SNN.**
>
> >This research cannot give full play to SNN strengths, like temporal processing capability, and will cause researchers to go astray.
>
> **R:** **Our Spikformer has competitive performance on neuromorphic datasets, demonstrating the temporal processing capability.** We hope our investigations can pave the way for further research on Transformer-based SNNs models.
>
> >The authors over-claimed their contributions.  The authors claim that they are the first to implement Transformer in SNNs.  Actually, there exist many spiking transformer works, e.g., [1-3].
>
> **R: Our Spikformer is the first architecture research in SNN-Transformer. The reference papers [1, 2] in your comments use ANN-Transformer** to process spike data although they have 'Spiking Transformer' in the title.
> Specifically, STNets [1] use a swin transformer, and temporal-spatial attention in ANNs to conduct object tracking.  Spike-T [2] adopts a variant of the swin transformer as the framework which is also an ANN.
>
> **Spiking Transformer Network [3] in your comments** trains a vision transformer and converts it to SNN, which is ANN2SNN rather than directly trained SNN. The floating-point position encoding,  and the vanilla self-attention which contains softmax function and float-point Query, Key, and Value, remain after the conversion. **Therefore, the network [3] is a hybrid architecture of SNN and ANN, and does not conform to the computation characteristic in SNN.**
>
> We add a discussion of the difference between these three works and Spikformer to the related work.

---

> > ### Author Response · Authors · 2022-11-11
> > **Response to Reviewer 6yty (2/3)**
> >
> > > The authors use floating point operations to add spikes, which defeats the purpose of SNNs.
> >
> > **R:** Two branches of spikes are added in an element-wise way (added directly) following SEW ResNet [2], which is a relatively popular SNN residual architecture[3, 4, 5]. After addition, the input to the subsequent block is indeed no longer binary spikes but multi-bit spikes.  Convolution layer and Linear layer are linear operations. Therefore, ignoring bias, computations in which multi-bit spikes are fed into the  layer can be decomposed into addition operations and multiplication is avoided, which is shown as
> > $$
> > {\rm Linear}(X_1 + X_2 + \cdots +X_{n}) =  {\rm Linear}(X_1) + {\rm Linear}(X_2) + \cdots + {\rm Linear}(X_n)
> > $$
> > $$
> > {\rm Conv}(X_1 + X_2 + \cdots +X_{n}) =  {\rm Conv}(X_1) + {\rm Conv}(X_2) + \cdots + {\rm Conv}(X_n)
> > $$
> > where $X_1,X_2,\cdots X_{n-1}$ is the skip connections of  binary spikes and $X_n$ is the current  binary spikes input.  Furthermore,  some neuromorphic chips can handle multi-bit spikes. For example, the Tianjic chip [6] supports 8-bit spike gradation, and Loihi can process up to 32-bit spike payload [7].
> >
> > >The authors are not familiar with the research progress of SNNs. There are many of the latest works, which are ignored by the authors. E.g. [4-7].
> >
> > **R:** Thanks for the reviewer's reminder, we refer to the above works in related work.
> >
> > >The comparison is unfair. As the network structure of the spiking transformer differs from SEW-ResNet-152, there is no doubt the performance will be better. I do not think it is a fair comparison. Please show the performance of the corresponding ANN.
> >
> > **R:** **Since there is no SNN-Transformer before, it is fair for us to make a direct comparison with SEW ResNet. With fewer parameters and similar theoretical energy consumption, Spikformer-8-512 (**73.38%**) outperforms 4.12% compared to SEW-ResNet-152 (**69.26%**) on ImageNet, which is a significant improvement.**
> >
> > Also, we provide a performance comparison with ANN-Transformer in the table above and add it to Tab. 2 in the manuscript.
> >
> > **On ImageNet**, ANN-Transformer-8-512 is 7.42% higher than Spikformer-8-512 (**80.80% VS. 73.38**), but the theoretical energy consumption is 3.31× of Spikformer-8-512 (**38.34 mJ VS. 11.58 mJ**).
> >
> > **On CIFAR10/100**, Spikformer-4-384 achieves 95.19% and77.86% and ANN-Transformer-4-384 achieves 96.73% and 81.02% after 300 epoch training.
> >
> > **At present ANN-transformer is better than our Spikformer, which is normal, but in the future, we will strive to narrow the performance gap.**
> >
> >
> > >Ignoring all that, the results are not convincing. The performance of the proposed method is not SOTA. The authors only chose favourable comparisons. For example, TET achieves 83.17% accuracy on CIFAR10-DVS dataset with 10 steps, while the proposed method gets 80.9% with 16 time-steps.
> >
> > **R:** The reason why TET outperforms Spikformer on CIFAR10DVS can be explained as follows:
> >
> > ① TET is not an architecture-based but a loss-based method that uses long epochs (300) and 9.27M VGGSNN, so we do not compare with it in the Tab. 4. We have also tried training Spikformer using TET's method, but with no performance improvement. The TET method may only apply to SpikeCNN.
> >
> > ② Transformer is a data-hungry model, it needs more data to train [8]. So, it does not do very well on small datasets. CIFAR10-DVS only contains 10K samples, which is even much smaller than CIFAR. This is the second reason to limit the performance of Spikformer on it.
> > **In the case of a large amount of training data, its ceiling is higher than ConvNet [9, 10, 11]. On ImageNet, we beat the TET-based network by a large margin.**

---

> > > ### Author Response · Authors · 2022-11-11
> > > **Response to Reviewer 6yty (3/3)**
> > >
> > > > The authors did not compare the training cost of the methods. Please add it in Table 2.
> > >
> > > **R:** **SNN mainly focuses on the theoretical SOPs and energy consumption at the stage of inference.** Because in theory, the SNN deployed on a hardware platform is pre-trained and only needs to perform inference tasks. For your comments, we provide training images per second on ImageNet on one V100 GPU for Spikformer and SpikCNN below.
> > > | Spikformer | Images/S|Acc (%)|SEW ResNet  |Images/S |Acc (%)|
> > > |-----------------|:--------:|:---------------:|:-----:|:-----:|:-----:|
> > > |Spikformer-8-384 |  134.814  | **70.24** | - | - | - |
> > > |Spikformer-6-512 |  121.870  |**72.46**  | SEW ResNet-34 | 187.494 | 67.04|
> > > |Spikformer-8-512 |   84.219  |**73.38**  | SEW ResNet-50 | 85.975  | 67.78|
> > > |Spikformer-10-512 |  75.214  |**73.68**  |SEW ResNet-101| 53.247 | 68.76|
> > > |Spikformer-8-768  | 49.948 | **74.81**| SEW ResNet-152| 37.678 | 69.26|
> > > **Table R2-2: Training speed on ImageNet.**
> > > In addition, the comparison of GPU memory usage when batch size is 2 is as follows:
> > >
> > > | Spikformer | Memory (M)|SEW ResNet  |Memory (M) |
> > > |-----------------|:--------:|:---------------:|:-----:|
> > > |Spikformer-8-384 |  4311    |   SEW ResNet-34 | 2803 |
> > > |Spikformer-6-512 |  3571    |   SEW ResNet-50 | 4253  |
> > > |Spikformer-8-512 |   4707    |   SEW ResNet-101| 5471 |
> > > |Spikformer-10-512 |  4869    |   SEW ResNet-152| 6873 |
> > > |Spikformer-8-768  | 5259 |
> > > **Table R2-3: Comparison of GPU Memory Usage .**
> > >
> > > > The ANN2SNN method achieves better performance with the increase of time-steps. How about spiking transformer? If you train the network with time-steps 4 on ImageNet, please show the performance of the network when the time-steps are 2,3,10,20,50.
> > >
> > > **R:** The ANN2SNN methods require large time steps to accurately approximate ReLU activation, which causes large latency [12]. While the advantage of directly trained SNNs is that they use fewer time steps. **It is not necessary to train Spikformer with 10, 20, and 50.
> > > We show in Tab. 5 that Spikformer can achieve higher accuracy as the increase of time steps (1, 2, 4, 6).** However, with the increase of the time step, the SNN performance of direct training will first increase and then saturate and even decay, which is also shown in the author's reply of SEW ResNet https://openreview.net/forum?id=6OoCDvFV4m&noteId=Cqa7AaMssZj. The smaller T means a lower latency, and is desirable. In Tab. 5, our network can achieve higher accuracy than SEW ResNet (T=4) with on time step (**70.14% VS. 69.26**).
> > >
> > > > Please compare SOPs and Power in Table 3 and 4.
> > >
> > > **R:** Most of the methods in Tab. 3 and 4 do not have reproducible codes and neuronal firing rates, which are needed to calculate theoretical SOPs and energy consumption, so they are not included in the Tabs. The results on ImageNet have demonstrated the superiority of our network over the current SNN network.

---

> > > > ### Author Response · Authors · 2022-11-11
> > > > **Reference**
> > > >
> > > > [1] Kaushik Roy, Akhilesh Jaiswal, and Priyadarshini Panda. Towards spike-based machine intelligence with neuromorphic computing. Nature, 575(7784):607–617, 2019.
> > > >
> > > > [2] Fang, W., Yu, Z., Chen, Y., Huang, T., Masquelier, T., & Tian, Y. (2021). Deep residual learning in spiking neural networks. _Advances in Neural Information Processing Systems (NeurIPS)_, 34, 21056-21069.
> > > >
> > > > [3] Hagenaars, J., Paredes-Vallés, F., & De Croon, G. (2021). Self-supervised learning of event-based optical flow with spiking neural networks. _Advances in Neural Information Processing Systems_, 34, 7167-7179.
> > > >
> > > > [4] Chen, Yanqi, et al. "State Transition of Dendritic Spines Improves Learning of Sparse Spiking Neural Networks." _International Conference on Machine Learning_. PMLR, 2022.
> > > >
> > > > [5] Vicente-Sola, A., Manna, D. L., Kirkland, P., Di Caterina, G., & Bihl, T. (2022). Keys to accurate feature extraction using residual spiking neural networks. _Neuromorphic Computing and Engineering_, 2(4), 044001.
> > > >
> > > > [6] Pei, Jing, et al. "Towards artificial general intelligence with hybrid Tianjic chip architecture." Nature 572.7767 (2019): 106-111.
> > > >
> > > > [7] Taking Neuromorphic Computing to the Next Level with Loihi 2. https://download.intel.com/newsroom/2021/new-technologies/neuromorphic-computing-loihi-2-brief.pdf
> > > >
> > > > [8] Liu, Y., Sangineto, E., Bi, W., Sebe, N., Lepri, B., & Nadai, M. (2021). Efficient training of visual transformers with small datasets. _Advances in Neural Information Processing Systems_, _34_, 23818-23830.
> > > >
> > > > [9] d’Ascoli, Stéphane, et al. "Convit: Improving vision transformers with soft convolutional inductive biases." _International Conference on Machine Learning_. PMLR, 2021.
> > > >
> > > > [10] Wang, Cong, et al. "Convolutional Embedding Makes Hierarchical Vision Transformer Stronger." _European Conference on Computer Vision. Springer_, Cham, 2022.
> > > >
> > > > [11] Chen, Zhengsu, et al. "Visformer: The vision-friendly transformer."  _In Proceedings of the IEEE/CVF International Conference on Computer Vision_. 2021.
> > > >
> > > > [12] Bing Han, Gopalakrishnan Srinivasan, and Kaushik Roy. Rmp-snn: Residual membrane potential neuron for enabling deeper high-accuracy and low-latency spiking neural network. In _Proceedings of the IEEE/CVF Conference on Computer Vision and Pattern Recognition (CVPR)_, pp. 13558–13567, 2020

---

> ### Author Response · Authors · 2022-11-17
> **Discussion Period Ending Soon**
>
> Dear reviewer 6yty,
>
> Thank you again for your valuable time and insightful comments. We have provided thorough responses to you and sincerely hope you can look through them and update the scores if your concerns have been resolved. We are also open to further discussion if the concerns have not been fully addressed. Please feel free to let us know if you still have any questions.
>
> Best regards!
>
> Authors

---

### Official Review · Reviewer_qGTH · 2022-10-25

**Confidence:** 4
**Correctness:** 4
**Technical Novelty And Significance:** 4
**Empirical Novelty And Significance:** 3
**Recommendation:** 6

**Clarity, Quality, Novelty And Reproducibility:**

Clarity; 8/10
Quality: 8/10
Novelty: 7/10
Reproducibility: Yes

**Strength And Weaknesses:**

Strength:

+ Clear paper structure, neat paper presentation.

+ Compared to spiking convolutional architectures the accuracy are higher.

Weakness:

- Seems like the method to alleviate the full precision multiplication is to add more LIF neurons.

- Can authors at least provide transfer learning results since high transferability is the core argument for artificial transformers? Also, more LIF may provide higher training latency in practice. Can the authors compare training GPUs between SpikeTransformer and SpikeCNN?


**Summary Of The Paper:**

This paper tackles the problem of adding spiking neurons to transformer architecture.

**Summary Of The Review:**

Based on the quality of this paper, I intend to give acceptance. If the concerns can be adequately addressed, I will increase my score.

---

> ### Author Response · Authors · 2022-11-12
> **Response to Reviewer qGTH**
>
> Thank you for your very detailed comments and suggestions for improvement. We would like to address your concerns and answer your questions in the following.
>
> >Seems like the method to alleviate the full precision multiplication is to add more LIF neurons.
>
> **R:**   We compare the number of LIF neuron layers and the numbers of Spikformer with SEW ResNet in Tab. R1-1. **It shows that Spikformer has fewer neuron layers and numbers than SEW ResNet but achieves higher accuracy** (Different neuron layers have different neuron numbers related to the feature dimension).
> There are indeed more neurons in each Spikformer block than in each SEW ResNet block because of self-attention characteristics, that is, the interaction and calculation between Query, Key, and Value, and the MLP block. Specifically, the Query, Key, and Value that interact **should be in spike form**, not only following the computational feature of SNN but also more straightforward to implement on hardware than in floating-point form. **We try our best to reduce the number of neurons while keeping Spikformer and SSA in line with the computational properties of SNN.**
> | Spikformer | SN Layers|SN Number (M)|Acc (%)|SEW ResNet  |SN Layers |SN Number (M)|Acc (%)|
> |-----------------|:--------------:|:---------------:|:-----:|:---:|:-----:|:-----:|:-----:|
> |Spikformer-8-384 |  33    |10.61  | 70.24| - | - | - | - |
> |Spikformer-6-512 |  47    |12.14  | **72.46** | SEW ResNet-34 |  33  |3.74 |67.04 |
> |Spikformer-8-512|    61  | 14.15 | **73.38** | SEW ResNet-50 | 49   | 11.11| 67.78|
> |Spikformer-10-512 |  75  | 16.16 | **73.68**  |SEW ResNet-101| 100  | 16.23| 68.76|
> |Spikformer-8-768 | 61 | 21.22| **74.81**  |  SEW ResNet-152| 151  | 22.55 |69.26 |
> **Table R1-1: Comparison of Neuron Numbers.**
>
> > Can authors at least provide transfer learning results since high transferability is the core argument for artificial transformers?
>
> **R:**  Thank you for your professional advice. We conduct the transfer learning on CIFAR based on the pre-trained Spikformer-4-384 and Spikformer-8-384/512, and analyse it in the Sec. 4.1 and Appendix. D.5 of the manuscript. The input size of CIFAR is $224 \times 224$, and the epoch number is 60.
> **As shown in Tab. R1-2, the experimental results show that Spikformer has good transfer ability.**
> | Directly Learning| CIFAR10 Acc (%)|CIFAR100 Acc (%)|
> |-----------------|:--------:|:---------------:|
> |Spikformer-4-384  |  95.19   | 77.86  |
> | **Transfer Learning**|
> |Spikformer-4-384  |  **95.54**    | **79.96**  |
> |Spikformer-8-384  |  **96.64**    | **82.09**  |
> |Spikformer-8-512 |  **97.03**    |  **83.83** |
> **Table R1-2: Transfer Learning on CIFAR.**
>
>
> > Also, more LIF may provide higher training latency in practice. Can the authors compare training GPUs between SpikeTransformer and SpikeCNN?
>
> **R: The training latency is not directly related to the number of LIF neurons.** The theoretical latency is related to
>
> ① the number of layers and synaptic operations (SOPs). The LIF neurons accumulate voltage and fire spikes, the computation can be ignored. There are no trainable parameters in LIF neurons.
>
> ② the simulation time step (T). As the simulation time increases, the latency becomes higher. As shown in Tab. 5, Spikformer-8-512 with T=1 achieve 70.14%, which is still higher than SEW ResNet-152 with T=4, demonstrating its potential for low latency.
>
> The training of Spikformer and SpikeCNN are both on eight Nvidia Tesla V100 (32G) GPUs. We provide training images per second on ImageNet on one V100 GPU for Spikformer and SpikCNN in Tab. R1-3.
> | Spikformer | Images/S|Acc (%)|SEW ResNet  |Images/S |Acc (%)|
> |-----------------|:--------:|:---------------:|-----|:-----:|:-----:|
> |Spikformer-8-384 |  134.814  | **70.24** | - | - | - |
> |Spikformer-6-512 |  121.870  |**72.46**  | SEW ResNet-34 | 187.494 | 67.04|
> |Spikformer-8-512 |   84.219  |**73.38**  | SEW ResNet-50 | 85.975  | 67.78|
> |Spikformer-10-512 |  75.214  |**73.68**  |SEW ResNet-101| 53.247 | 68.76|
> |Spikformer-8-768  | 49.948 | **74.81**| SEW ResNet-152| 37.678 | 69.26|
> **Table R1-3: Training speed on ImageNet.**
>
> In addition, the comparison of GPU memory usage when batch size is 2 is as follows:
> | Spikformer | Memory (M)|SEW ResNet  |Memory (M) |
> |-----------------|:--------:|---------------|:-----:|
> |Spikformer-8-384 |  4311    |   SEW ResNet-34 | 2803 |
> |Spikformer-6-512 |  3571    |   SEW ResNet-50 | 4253  |
> |Spikformer-8-512 |   4707    |   SEW ResNet-101| 5471 |
> |Spikformer-10-512 |  4869    |   SEW ResNet-152| 6873 |
> |Spikformer-8-768  | 5259 |
> **Table R1-4: Comparison of GPU Memory Usage.**

---

> ### Author Response · Authors · 2022-11-17
> **Discussion Period Ending Soon**
>
> Dear reviewer qGTH,
>
> Thank you again for your valuable time and insightful comments. We have provided thorough responses to you and sincerely hope you can look through them and update the scores if your concerns have been resolved. We are also open to further discussion if the concerns have not been fully addressed. Please feel free to let us know if you still have any questions.
>
> Best regards!
>
> Authors

---

> > ### Comment · Reviewer_qGTH · 2022-11-27
> > **Appreciate it.**
> >
> > Thank you for your detailed response. I'd like to keep my original score not because I think this paper is only weak accept, but because I think the current scores rated by other reviewers are enough for this draft. I'd like to see this paper published on ICLR.

---

### Author Response · Authors · 2022-11-14
**To all reviewers**

We thank all reviewers for their time and helpful comments. We have responded to all reviewers' comments and uploaded a revised version of our manuscript with all changes marked in blue.

We respectfully point out that the claim of Reviewer 6yty, which is
>The authors over-claimed their contributions. The authors claim that they are the first to implement Transformer in SNNs. Actually, there exist many spiking transformer works, e.g., [1-3].

is inaccurate .  **The works in literature 1, 2, and 3 in your comments are not SNN-Transformers.**

Specifically, although there is 'Spiking Transformer' in the title of the reference [1, 2] in your comments, actually, **ANN-Transformers** are used to process spike data in these two papers. While **Spiking Transformer Network [3] in your comments is a hybrid Transformer of SNN and ANN**, which trains a transformer and converts it to SNN. The floating-point position encoding and the vanilla self-attention which contains softmax function and floating-point Query, Key, and Value, remain after the conversion. Therefore, the network does not conform to the computation characteristic in SNN.

**Our Spikformer is indeed the first directly training SNN-Transformer.** We argue that introducing the Transformer architecture to SNN is of great significance, and Spikformer, as the first SNN-Transformer backbone, can enrich SNN's architecture and improve SNN's performance.


Thanks,
Paper553 Authors

---

### Decision · Program_Chairs · 2023-01-20

**Decision:**

Accept: poster

**Justification For Why Not Higher Score:**

The significance of this work is limited to pushing the SOTA in spiking neural networks.

**Justification For Why Not Lower Score:**

There is general support for this paper to be accepted except for one reviewer. My read is that the authors addressed relatively well the criticisms but I was not able to get any of the reviewers to discuss...

**Metareview: Summary, Strengths And Weaknesses:**

This paper describes an approximation of transformer networks that can be implemented by spiking neural networks. The main and only contribution of the paper is to simplify the self-attention module of the transformer by removing the softmax function to enable a spiking network approximation. Hence the technical contribution appears quite limited and the paper's main contribution is in the fact that it pushes the SOTA for spiking models. But as expected, while the resulting spiking transformer outperforms spiking CNNs on imagenet and other image datasets used to evaluate spiking neural networks, all spiking networks fair far below non-spiking SOTA models. Hence, this paper will only generate interest from people interested in spiking neural networks (which is a rapidly growing but remains somewhat of a niche for ML).

**Note From Pc:**

if the above contains the word "oral" or "spotlight" please see: "oral" presentation means -> notable-top-5% and "spotlight" means -> notable-top-25%. As stated in our emails, we are disassociating presentation type from AC recommendations